# Comparison of size distribution and electrical particle sensor measurement methods for particle lung deposited surface area (LDSA[al]) in ambient measurements with varying conditions

Teemu Lepistö[1], Henna Lintusaari[1], Laura Salo[1], Ville Silvonen[1], Luis M.F. Barreira[2], Jussi Hoivala[1], Lassi Markkula[1], Jarkko V. Niemi[3], Jakub Ondracek[4], Kimmo Teinilä[2], Hanna E. Manninen[3], Sanna Saarikoski[2], Hilkka Timonen[2], Miikka Dal Maso[1], Topi Rönkkö[1]

[1]Aerosol Physics Laboratory, Physics Unit, Faculty of Engineering and Natural Sciences, Tampere University, Tampere, 33014, Finland
[2]Atmospheric Composition Research, Finnish Meteorological Institute, Helsinki, 00101, Finland
[3]Helsinki Region Environmental Services Authority HSY, Helsinki, 00066, Finland
[4]Research Group of Aerosol Chemistry and Physics, ICPF CAS, Prague, 165 00, Czechia

*Correspondence to*: Teemu Lepistö (teemu@lepisto@tuni.fi)

**Abstract.** It has become evident that additional metrics along the particle mass concentration together with dense air quality monitoring networks within cities are needed to understand the most efficient ways to tackle the health burden of particulate pollution. Particle lung deposited surface area (LDSA[al]) is a metric to estimate particle exposure in the lung alveoli, and it has gained interest as a parameter for air quality monitoring as it is relatively easy and cost-efficient to measure with electrical particle sensors. Also, various studies have indicated its potential as a health-relevant metric. In addition to the electrical particle sensors, LDSA[al] can be measured with various size distribution methods. However, different LDSA[al] measurement methods have fundamental differences in their operation principles e.g., related to the measurement size ranges, size-classification or conversion from the originally measured quantity into LDSA[al]. It is not well understood how these differences affect the accuracy of the measurement in ambient conditions where especially the particle effective density and hygroscopicity can considerably change the particle lung deposition efficiencies. In this study, the electrical particle sensor measurement (Partector) and two size distribution approaches (ELPI+, DMPS/SMPS) were compared in road traffic environments with different environmental conditions in Helsinki and Prague. The results were compared by utilising general assumptions of LDSA[al] measurement (spherical hydrophobic particles with the standard density) and by evaluating the effects of the particle effective density and hygroscopicity. Additionally, the Partector and ELPI+ were compared in various urban environments near road traffic, airport, river traffic and residential wood combustion. The results show that comparison of different LDSA[al] measurement methods can be complicated in ambient measurements. The challenges were especially related to the accumulation mode particles roughly larger than 200–400 nm for which the dominant deposition mechanism in the lung changes from diffusion to impaction, and the particle effective density and hygroscopicity tend to increase. On the other hand, the results suggest that the differences between the methods are reasonably low when considering only ultrafine and soot

particles, which have effective density closer to the standard ($1.0$ g/cm$^3$) and are more hydrophobic, highlighting the suitability

of LDSA[al] as a monitored metric when estimating spatial differences in the particulate pollution within cities.

## 1 Introduction

Even though particulate pollution is known to be harmful for human health, it is still not comprehensively understood what the main mechanisms behind the negative health effects are, nor how to monitor and regulate the health-relevant particulate emissions most efficiently. Since a study by Dockery et al. (1992), health effects of particulate pollution have been associated

especially with fine particulate matter (PM$_{2.5}$), i.e., the mass concentration of particles smaller than 2.5 µm. PM$_{2.5}$ is also the most widely used metric for air quality monitoring, regulations, and recommendations for ambient particles. However, various studies have indicated that monitoring of only PM$_{2.5}$ is not enough in terms of the negative health effects. For example, epidemiological studies have suggested that the dose-response function between PM$_{2.5}$ and the health effects is not linear, and PM$_{2.5}$ seems to be relatively more harmful in lowly polluted regions compared to highly polluted ones (e.g., Pineault et al.

2016, Vodonos et al. 2018, Strak et al. 2021). Furthermore, in a study by Daellenbach et al. (2020), it was observed that the main sources of particulate mass (PM$_{10}$, particles smaller than 10 µm) and particle oxidative potential (OP) are not the same in different locations across Europe. Also, PM$_{2.5}$ toxicity is suggested to be considerably dependent on the emission source and composition (e.g., Jia et al. 2017, Park et al. 2018, Sidwell et al. 2022). All these findings highlight the need for other methods, metrics, and point-of-views along with the particle mass for regulating and monitoring the particulate pollution to

better tackle the adverse health effects of air pollution.
Different particle physical and chemical characteristics likely have a major role in the varying PM$_{2.5}$ health effects. The chemical composition of particles affects the toxicity and OP of particles as indicated e.g., by Park et al. (2018) and Daellenbach et al. (2020). Also, the particle size affects the toxicity, which has been suggested to increase as a function of decreasing particle size and increasing surface area concentration (Oberdorster 2005, Schmid and Stoeger 2016, Hakkarainen

et al. 2022). Furthermore, the particle size affects the particle respiratory tract deposition, and especially ultrafine particles, i.e., particles smaller than 100 nm, deposit efficiently in the lung alveoli (ICRP 1994). The health effects of ultrafine particles are not properly recognised yet, but they have been linked e.g., to diabetes and myocardial infarction as well as with changes in inflammatory status and cardiovascular conditions (Ohlwein et al. 2019, Vallabani et al. 2023). By measuring only PM$_{2.5}$, the differences in particle composition or in ultrafine particle concentrations cannot properly be detected (e.g., de Jesus et al.

2019, Chen et al. 2022). Thus, equal PM$_{2.5}$ concentrations in different environments can consist of varying combinations of physical and chemical properties of particles, which likely influences the health effects. Moreover, the composition and ultrafine particle concentrations are typically strongly dependent on the nearby pollution sources which emphasises the need for dense air quality measurement networks (see e.g., Kuula et al. 2020, Edebeli et al. 2023) especially in cities to better observe and recognise the aerosol that people are exposed to in different locations. For example, it has been suggested that

within-city $PM_{2.5}$ dose-response gradients are steeper than between-city gradients, emphasising the role of near-source exposure (e.g., to traffic) in terms of adverse health effects of particles (Segersson et al. 2021).

As the current scientific evidence of particle health effects highlights the need for more detailed particle characterisation as well as for more dense air quality monitoring network within cities, it is crucial to understand what properties of particles should be monitored. Even though the particle chemical composition and OP are likely key factors in the health effects, their

utilisation in monitoring purposes is practically challenging due to the expensive and complicated instrumentation (e.g., Onasch et al. 2012). The World Health Organization (WHO) has recommended the measurement of ultrafine particles and black carbon (BC) in the revised air quality guidelines as good practice statements (WHO 2021). Also, the measurement of particle lung deposited surface area (LDSA[al]) is an interesting option for monitoring measurements. LDSA[al] measures the surface area concentration of particles that deposit in the lung alveoli where the interaction between the pulmonary circulation

and the respiration occurs. Particles entering the lung alveoli can therefore possibly end up in the blood and other organs like the brain (Heusinkveld et al. 2016). The association between the health effects and LDSA[al] are not completely known but e.g., studies by Aguilera et al. (2016) and Patel et al. (2018) indicate that LDSA[al] has stronger associations with subclinical atherosclerosis and reduced lung function than the particle mass, respectively. Also, LDSA[al] concentration as a function of $PM_{2.5}$ has been found to have similar behaviour as the $PM_{2.5}$ dose-response function, which highlights the potential of the

metric in terms of the health effects (Lepistö et al. 2023). It's worth noting that, in different studies, LDSA can also be referred when considering other respiratory tract regions than the alveoli (e.g. Liu et al. 2023). Here, the notation LDSA[al] is used to clarify that only alveolar deposition is considered in this study (see Lepistö et al. 2023).

For air quality monitoring, LDSA[al] is a convenient metric (e.g., Kuula et al. 2020, Edebeli et al. 2023) as it is reasonably easy to measure with electrical particle sensors such as the Partector (Fierz et al. 2014), nanoparticle surface area monitor (NSAM,

Shin et al. 2007), Aerasense MP (Marra et al. 2019) and Pegasor PPS-M (Järvinen et al. 2015). In addition, LDSA[al] is strongly affected by local emissions of ultrafine particles (Liu et al. 2023, Lepistö et al. 2023) and BC (e.g., Reche et al. 2015, Kuula et al. 2020, Lepistö et al. 2022) which both have been indicated to be important health-relevant parameters (e.g., Janssen et al. 2011, WHO 2021). Therefore, the electrical particle sensor measurements of LDSA[al] could provide a relatively easy and cost-efficient method to monitor local particulate pollution with a dense monitoring network within cities.

In addition to the electrical particle sensors, LDSA[al] can be measured with a size distribution -based measurement where the obtained size distributions are weighted with the particle lung deposition efficiency function. For example, the scanning/differential mobility particle sizer (SMPS/DMPS) and the electrical low pressure impactor (ELPI+) have been utilised in LDSA[al] measurements (e.g., Lepistö et al. 2020, Teinilä et al. 2022, Liu et al. 2023, Chen et al. 2023, Lepistö et al. 2023). These three approaches have major differences in their fundamental operation principles. For example, the electrical

methods (sensors and the ELPI+) determine the surface area based on the electric charge of particles after a diffusion charger (proportional to particle size) whereas, with the SMPS/DMPS, the surface area is determined based on the particle number size distribution, e.g., by assuming spherical particles. Also, the electrical particle sensors measure LDSA[al] by assuming certain particle size in the calibration, and they measure LDSA[al] with a reasonable accuracy only up to roughly 400 nm (Todea et al.

2015). Furthermore, with the size distribution methods, particle size classification depends on different particle concepts with the ELPI+ (aerodynamic diameter) and the SMPS/DMPS (mobility equivalent diameter), and, hence, assumptions of the particle effective density cause uncertainty between the methods if additional instrumentation is not available to determine the effective density accurately. Even though these limitations of the instruments are generally well known, the differences in the reported LDSA[al] concentrations or size distributions with the different methods in varying ambient conditions are poorly understood.

Also, a better understanding of how well the different methods actually measure the particle lung deposition is needed. As mentioned, the size distribution methods need assumptions of particle effective density which causes uncertainty in the devices' operation. However, the effective density also affects the particle lung deposition efficiency which increases the vulnerability to errors even more in LDSA[al] measurement. Moreover, the hygroscopic growth of particles in the lungs is often neglected in LDSA[al] measurements as it requires detailed information of the particle composition. Both the effective density and the hygroscopic growth can considerably change the particle lung deposition efficiencies (e.g., Löndahl et al. 2014, Vu et al. 2015, Lizonova et al. 2024). These factors are however practically challenging to consider in air quality monitoring measurements as they require additional instrumentation which cannot be considered as a realistic approach with dense air quality monitoring networks. Also, with the electrical particle sensor measurement, these factors cannot be taken into account. In addition, particle lung deposition efficiencies are individual and dependent on the human anatomy and the breathing pattern, and, thus, approximations of the exposed population are always needed.

To understand the suitability of LDSA[al] measurement in air quality monitoring measurements, first, it is important to know how the different measurement methods compare with each other. Second, it is crucial to know how well the different methods, with their typical measurement assumptions, actually measure the particle lung deposition, and whether there are variations in the performances of devices in different locations and conditions.

In this study, LDSA[al] measurement with an electrical particle sensor (Partector), an ELPI+ and a mobility particle sizer (DMPS or SMPS) were compared at road traffic sites with varying conditions in Helsinki (Finland) and Prague (Czechia). The aim was to understand how well the different measurement methods compare with each other and how vulnerable they are to errors in the estimated lung deposition due to the assumptions related to the particle effective density and hygroscopicity. Furthermore, reported LDSA[al] concentrations with the Partector and ELPI+ were compared in additional measurements in Tampere (Finland) and Düsseldorf (Germany), including road and river traffic, airport, and residential wood combustion influenced aerosol to better understand the location-dependent performance of the electrical particle sensor measurement. Ultimately, the study aimed to evaluate the suitability of LDSA[al] as a metric for air quality monitoring measurements in general, and to help the interpretation of previous and future works on ambient LDSA[al] measurements using different methodologies.

## 2 Materials and methods

The comparisons with the three different LDSA[al] measurement methods (Partector, ELPI+, DMPS/SMPS) were conducted in road traffic environments in Helsinki (Finland) and Prague (Czechia). The DMPS was used to measure LDSA[al] in Helsinki and the SMPS was used in Prague. Furthermore, the Partector and ELPI+ were also compared in different urban environments in Tampere (Finland) and Düsseldorf (Germany), including road and river traffic sites, an airport and a site affected by residential wood combustion near a detached-housing area. Maps of the measurement locations are collected in the

Supplementary Information (Fig. S1-4).

### 2.1 Instrumentation

The Partector (Naneos GmbH, Fierz et al. 2014) represents the electrical particle sensor measurement method for LDSA[al] which is based on detecting the electric current caused by the sampled particles after a diffusion charger. The diffusion charging efficiency is determined as a multiplication of the number of elementary charges of a particle after charging ($n$), and the

probability of a particle to penetrate through the charger ($P$). The product, $Pn$, is dependent on the particle mobility equivalent size with an exponent varying typically between 1.1– 1.9  (Dhaniyala et al. 2011, Järvinen et al. 2014). Due to a lucky coincidence, the charger efficiency correlates reasonably well with LDSA[al] of a single particle in a size range roughly from 20 nm to 400 nm, which can, however, be altered slightly by adjusting the ion trap voltage of the charger (Fissan et al. 2006). The Partector first charges the sampled particles in a diffusion charger and then converts the detected electric current caused by the

sampled particles into LDSA[al] concentration with a single calibration factor. The chosen calibration factor is the response coefficient between the electric current and LDSA[al] at 100 nm, which typically is close to the peak size of LDSA[al] size distributions in urban environments (Fierz et al. 2014). Similar approach of converting diffusion charged current into LDSA[al] is utilised also with other sensor-type instruments such as the nanoparticle surface area monitor (NSAM, Shin et al. 2007), Aerasense MP (Marra et al. 2019) and Pegasor PPS-M (Järvinen et al. 2015). The main advantage of the method is that it

enables measurement with low-maintenance handheld devices. Also, LDSA[al] can be determined with 1 s time-resolution. On the other hand, the method is reasonably accurate only for particles roughly from 20 nm to 400 nm and, thus, accurate performance of the device requires sampled particles to be in the certain size range.

The ELPI+ (Dekati Oyj, Keskinen et al. 1992, Järvinen et al. 2014) is a particle size distribution measurement device which utilises a 14-staged cascade impactor to classify the sampled particles according to their aerodynamic size. Before the size

classification, the sampled particles are charged in a diffusion charger, similarly as with the electrical sensors. The electric current caused by the collected particles in each impactor stage is measured with electrometers and can then be converted, e.g., to particle number, mass or LDSA[al] (Lepistö et al. 2020). Each impactor stage has its own conversion factors into the wanted quantity depending on the particle size. The 14 stages enable measurement from 6 nm up to 10 µm, and the time-resolution of the measurement is 1 s. As the particle charge after the diffusion charger, and, therefore, the measured electric current, is

dependent on the particle mobility equivalent diameter, and the size classification is dependent on the aerodynamic diameter,

the ELPI+ measurement requires estimation of the particle effective density to estimate the average electric current caused by a single particle collected onto a impactor stage and to convert the measured current to other quantities accurately.

The DMPS and SMPS both share the same operation principle of measuring particle size distributions by utilising a combination of a differential mobility analyser (DMA) and a condensation particle counter (CPC). First, the DMA is used to select only certain sized particles to remain in the sample flow according to their electrical mobility. Then, the remaining sample is measured with the CPC, hence, the number concentration of particles in a certain size range can be determined. By adjusting the DMA parameters, the size range of the measured particles can be changed, enabling the measurement of particle number size distribution. Then, the obtained number size distribution can be weighted with the particle lung deposition function and, thus, the LDSA[al] concentration and size distribution can be measured. Here, the utilised lung deposition function (similarly as with the ELPI+ calibration) was based on the ICRP-model with averaged data for males and females at three physical activity levels: sitting, light exercise and heavy exercise (ICRP 1994, Hinds 1999). The ICRP-model is a semi-empirical regional compartment lung-deposition model which considers the human respiratory tract as a series of filters and utilises measured data with human volunteers, The parameters used in the ICRP-model are provided in Table S1. The size range and resolution as well as the time resolution of the measurement depends on the chosen DMA parameters. In general, the time resolution of the method is lower than with the electrical methods. In this study, the DMPS and SMPS measured particles from 10 nm to 800 nm and from 10 nm to 500 nm with time resolutions of 520 s and 300 s, respectively. The DMPS system (Helsinki) consisted of a Vienna type DMA and A20 CPC (Airmodus). The SMPS system (Prague) consisted of EC 3080, DMA 3081 and CPC 3772 (all TSI).

In addition to LDSA[al] measurement devices, an AE33 aethalometer (Magee Scientific, Drinovec et al. 2015) was used to measure black carbon (BC) concentration and Teledyne Model T201 was used to measure nitric oxide (NO) concentration during the measurements in Helsinki and Prague.

### 2.1.1 Differences and challenges with the methods in LDSA[al] measurement

As none of the described methods directly measures the particle lung deposition, LDSA[al] is determined with conversion factors from the measured quantity. Generally, the conversion factors into LDSA[al] are determined by assuming the measured particles to be spherical with the standard density (1.0 g cm[-3]), and that the particles do not grow in the human lungs due to hygroscopicity. With the Partector and ELPI+ the measured electric current is converted into LDSA[al] whereas, with the DMPS and SMPS, LDSA[al] is converted from the measured number size distribution.

With the Partector, despite the reasonably good correlation between the electric current and LDSA[al] with 20–400 nm particles, the needed conversion factor is dependent on the assumed particle size distribution in the calibration, hence, the accuracy is ±30 % in this size range (Todea et al. 2015). LDSA[al] of particles smaller than 20 nm can generally be assumed to be low due to the small particle size, but LDSA[al] of particles larger than 400 nm can greatly be underestimated with the method. For example, in highly polluted environments, the regional aerosol and the accumulation mode particles dominate the particle size distribution and, thus, particles larger than 400 nm can have a significant effect on LDSA[al] (Salo et al. 2021; Lepistö et al.

2023). Therefore, the performance of the Partector may considerably vary depending on the dominant pollution source and regional aerosol concentrations.

With the size distribution methods, the size-dependence of the conversion factors can be taken into account, and, in principle, varying particle size distributions should not affect the measurement accuracy. However, there are other fundamental challenges with the methods as the ELPI+ measures the size distributions according to the aerodynamic size whereas the DMPS/SMPS measure them based on the mobility equivalent size. The particle lung deposition is driven by the diffusion with smaller particles (roughly < 0.1 µm) which is dependent on the mobility equivalent size whereas larger particles (> 0.5 µm) deposit due to impaction and sedimentation which are depended on the aerodynamic size (Hofmann et al. 2011). Therefore, the particle effective density not only affects the comparability of the devices, but it also affects the particle lung deposition efficiencies (Löndahl et al. 2014, Lizonova et al. 2024) and, hence, the accuracy of the size distribution methods in terms of LDSA[al]. Also, it is worth noting that the size ranges of the size distribution measurements vary depending on the study and the device (also in this study), and the studied size range is not typically considered when reporting LDSA[al] concentration, even though it can considerably affect the results.

The role of the effective density is also important when considering the conversion from electric current or particle number into LDSA[al]. As mentioned, in calculation, it is generally assumed that the particles are spherical but, in reality, ambient particles can have agglomerated and non-spherical structures. On the other hand, the error due to the spherical particle assumption is likely less significant with the electric current measurement as the diffusion charged current is proportional to particle size and shape whereas the particle number is not. Therefore, measurement of the electric current may better consider the non-spherical structure of particles in terms of the surface area which, on the other hand, increases the uncertainty when comparing the different methods (see also Chang et al. 2022, Chen et al. 2023).

Despite the varying operation principles, the different LDSA[al] measurement methods have shown good agreement with each other in laboratory measurements (e.g., Leavay et al. 2013, Todea et al. 2017, Lepistö et al 2020), showing that, in principle, all the methods are suitable to measure LDSA[al]. However, the comparability of the methods in varying ambient conditions with varying particle characteristics is not well known. For example, Kuula et al. (2019) reported good agreement between the DMPS and various electrical particle sensors at a road traffic site in Helsinki, whereas Chen et al. (2023) observed roughly 1.5 times higher LDSA[al] concentrations with a NSAM than with a SMPS at a road traffic site in Taiwan. In addition, it should be noted that it is not well known how well the results represent the actual particle lung deposition as, for example, the particle hygroscopicity can considerably change the particle lung deposition efficiency along with the effective density (Vu et al. 2015). On the other hand, the neglected hygroscopic growth of particles, together with the standard density assumption, are often the only reasonable options for monitoring measurements as the consideration of these parameters require additional sophisticated instrumentation. Also, in principle, particle lung deposition efficiencies are individual and dependent on the human anatomy and the breathing pattern. Thus, the utilised lung deposition efficiency functions in device calibrations are always approximations, and the chosen approaches may vary with different instruments. Thus, in addition to the uncertainties between

the different operation principles of the methods, LDSA[al] measurements also have uncertainty in the estimation of the actual particle lung deposition.

## 2.2 Measurement campaigns

**2.2.1 Helsinki**

In Helsinki, the measurements were conducted during daytime in Mäkelänkatu street canyon in the city centre (60.1963 N, 24.9523 E) on 18 January – 16 February 2022. The measurements were done on a kerbside both in an air quality monitoring supersite operated by Helsinki Region Environmental Services Authority (HSY) and right next to the site in the Aerosol and trace-gas mobile laboratory (ATMo-Lab). The ATMo-Lab is a van which takes the sample above the windshield at the height

of 2.2 metres and then divides the sample for the instruments located in the back-end of the vehicle (see, e.g., Lepistö et al. 2023). The street canyon includes three driving lanes to both directions and two tram lines as well as traffic-light junctions. In general, street canyons weaken the dispersion of the road traffic emissions (see exact characterisation of the same street canyon by Barreira et al. 2021). The ELPI+, Partector, AE33 and T201 measured in the ATMo-Lab and the DMPS measured in the supersite. The measurements were carried out daily between 6.30 am and 7.30 pm but the ATMo-Lab was also utilised in

driving measurements during the measurement hours which are not considered in the analysis of this study. Detailed description of the measurements is provided by Teinilä et al. (2024).

The conditions during the measurements were typical winter-time conditions in Helsinki, the average (min–max) temperature, relative humidity and wind speed being -1.5 (-11.1–2.9) °C, 88 (58–100)% and 4.8 (0.6–11.4) m s$^{-1}$, respectively (Teinilä et al. 2024). On 31 January – 5 February an episode of cold weather occurred (temperature below -5 °C) which reduced the

dilution and dispersion of pollutants, highlighting the contribution of local emissions within the city. This period is referred here as an inversion episode. Also, on 13 February a long range transported (LRT) pollution episode started which lasted until the end of the measurements. During the LRT-episode, air masses in Helsinki had travelled through Central and Eastern Europe (Teinilä et al. 2024). This period is referred as a LRT-episode in this study.

**2.2.2 Prague**

The measurements in Prague were carried out during five days and one night on 25 March – 3 April 2022 next to a two-lane street with two tramlines near a train station in Vršovice (50.0664 N, 14.4462 E). In comparison with the Helsinki street canyon site, the measurement site was in an open environment in a preschool yard behind a fence, which limited the direct effects from the nearby traffic. As in Helsinki, the measurements were conducted in a monitoring station and in the ATMo-Lab next to the station. The same ELPI+, Partector, AE33 and T201 units as in Helsinki were installed in the ATMo-Lab whereas the

SMPS measured in the monitoring station. During the studied period, the ATMo-Lab was also utilised in driving measurements and in another measurement location which are not considered in this study. The average (min–max) temperature, relative

humidity and wind speed were 8.1 (0.5–19.7) °C, 65 (24–96) % and 2.8 (0.3–6.1) m s$^{-1}$, respectively (data provided by Czech Hydrometeorological Institute).

### 2.2.3 Additional measurements in Tampere and Düsseldorf

Measurements with the ATMo-Lab, equipped with the same ELPI+ and Partector units, were conducted also in Tampere and Düsseldorf. In Tampere, the measurements were done in an industrial area which is located next to a train yard, a highway, and detached housing areas on 30 November – 20 December 2021. In general, the main source of particles in the measurement site was the road traffic but during an inversion episode (7–9 December), emissions of the nearby detached housing areas dominated the sampled aerosol. Detailed description of the Tampere measurements is provided by Silvonen et al. (2023). In

Düsseldorf, the measurements were carried out in an urban traffic site, on a highway, near an airport and on a riverside of Rhine during 8–23 March 2022. The detailed descriptions of the Düsseldorf measurements as well as the ELPI+ LDSA$^{al}$ results are provided by Lepistö et al. (2023). The data from Düsseldorf were utilised with similar criteria as in the corresponding publication.

### 2.3 Data processing and analysis

Only the data which were measured when all the LDSA$^{al}$ instruments operated in certain environment were considered in the analysis. The presented results are based on the geometric mean of the observed concentrations. With the ELPI+, Partector, AE33 and T201, the data were however first changed to 1 min resolution with an arithmetic mean to reduce noise. With the ELPI+, the upper limit of the measurement size range was 2.5 µm. In addition to LDSA$^{al}$, the ELPI+ was also used to measure particle number (PN) and PM$_{2.5}$ concentrations by integrating the obtained particle number and mass size distributions. In Fig.

S23 and S25, LDSA$^{al}$ concentration of particles smaller than 400 nm with the ELPI+ was determined by considering the data from impactor stages 1–7 which correspond to 50 % cut-off diameters starting from 6 nm to 383 nm. The data from Helsinki and Prague are divided into four categories: 1. Measured data in Helsinki ignoring the episodes (Helsinki: No episode), 2. Measured data during the inversion episode in Helsinki (Helsinki: Inversion), 3. Measured data during the LRT-episode in Helsinki (Helsinki: LRT), and 4. All measured data in Prague (Prague: All). The data from Tampere were divided based on

the conditions (all data without the inversion episode and during the inversion), and the data from Düsseldorf were divided based on the measurement location: 1. Tampere: No episode, 2. Tampere: Inversion, 3. Düsseldorf: Urban traffic, 4. Düsseldorf: Highway, 5. Düsseldorf: Airport and 6. Düsseldorf: River.

With the ELPI+, SMPS and DMPS, LDSA$^{al}$ concentrations and size distributions were determined with three different methods. First, by utilising the general assumptions, i.e., particles have standard effective densities ($\rho_{eff} = 1$ g cm$^{-3}$), and they

do not grow in the human lungs due to hygroscopicity. Second, sensitivity analysis of the LDSA$^{al}$ calculation was done by correcting the results with an estimated effective density value but not with the hygroscopic growth. Third, the calculation was corrected with estimations of both particle effective density and the hygroscopic growth. With the Partector, these corrections cannot be applied in the results.

The particle effective density for the sensitivity analysis was estimated by comparing the peak sizes of the surface area size distributions of the ELPI+ to those of the DMPS or SMPS. The relationship between the aerodynamic ($d_a$) and the mobility equivalent diameter ($d_m$) is

$$d_m = d_a \sqrt{\frac{C_c(d_a)}{\rho_{\text{eff}} C_c(d_m)}}, \qquad (1)$$

where $C_c$ is the Cunningham slip correction factor. Hence, the average effective density of particles can be estimated by matching the peak sizes of the surface area size distributions (Fig. S5-6). The effective density correction (Lepistö et al. 2020) was done by utilising one estimation for the effective density for each studied location, based on the average size distributions of all the measured data in the certain location. The correction calculates the density corrected deposition function by considering inertial deposition (aerodynamic diameter) and diffusional deposition (mobility equivalent diameter) separately (see ICRP 1994, Lepistö et al. 2020). The surface area size distribution was chosen for the comparison as it was considered to be the most relevant unit in terms of LDSA[al]. With this approach, the average effective density in terms of particles contributing to LDSA[al] can be approximated for sensitivity analysis, but it should be noted that, in reality, the effective density depends on the particle size and has temporal variation. On the other hand, in monitoring measurements, it is not generally possible to monitor the effective density nor its size-dependence with high time resolution, and a representative value which applies for all the observed data must be chosen, supporting the chosen approach. Also, with the ELPI+, data analysis with size-dependent effective density is not straightforward due to the cascade impactor measurement. Furthermore, in this study, it was not possible to determine the temporal variation of the effective densities reliably due to the relatively slow SMPS or DMPS measurement. The effect of particle hygroscopic growth on the particle lung deposition functions was estimated according to the study by Vu et al. (2015) for road traffic environments. The method utilises data of size-dependent hygroscopic growth ratios of particles observed in road traffic environments which are then taken into account in the lung deposition efficiency calculations by adjusting the particle size. The growth rate is calculated by assuming relative humidity of 99.5 % in the human lung. It should be noted, that the hygroscopicity correction only changes the estimated lung deposition efficiency of particles, not the initial size distribution or the surface area of the inhaled particles. This comparison should be considered as an indicative representation of the effects of particle hygroscopic growth in terms of LDSA[al] measurements as the particle hygroscopicity is dependent on the particle composition which was not analysed in this study. The approach, however, provides valuable information of the accuracy of the studied methods in terms of actual particle lung deposition as the particle hygroscopicity has generally been neglected in previous LDSA[al] studies. The utilised hygroscopicity corrected lung deposition function (Vu et al. 2015) and the non-corrected one for spherical particles with standard density with the ELPI+ and the DMPS/SMPS data are shown in Fig. S7.

**3 Results and discussion**

**3.1 General overview of the measurements**

The average measured PN, PM$_{2.5}$, NO, and BC concentrations in Helsinki and Prague during the studied periods are collected in Table 1. Also, the estimated average particle effective densities for the sensitivity analysis are shown in the table. In general, the contribution of the nearby road traffic was clearer in Helsinki than in Prague due to the shorter distance from the passing vehicles to the measurement site, partly explaining the relatively higher average PN, NO and BC concentrations compared to PM$_{2.5}$. In Helsinki, PM$_{2.5}$ concentration was mainly low (average of 3.4 µg m$^{-3}$ without the episodes), indicating low regional

pollution in general. In Prague, the average PM$_{2.5}$ was considerably higher (20.2 µg m$^{-3}$) which was mainly related to accumulation mode particles and regional aerosol even though the higher NO also suggests effects from the traffic within the city. In Helsinki, PM$_{2.5}$ concentration increased during the inversion- and LRT-episodes. During the inversion episode also PN, NO and BC concentrations increased considerably which indicates local contribution. In addition to road traffic, higher BC during the inversion episode indicates effects of residential wood combustion, which is typical emission source in Finland

during winter (e.g. Teinilä et al. 2022). During the LRT-episode, increases with PN, NO and BC were less significant than during the inversion episode when considering also the higher PM$_{2.5}$, supporting the idea of long range transported pollution. Histograms of the measured concentrations are provided in Fig. S8-11.

**Table 1: Average measured PN, PM$_{2.5}$, NO and BC concentrations in Helsinki and Prague. Also, the estimation of the average**
**particle effective density ($\rho_{eff}$) for sensitivity analysis is shown. *The estimated particle effective density in Prague could have been higher based on only the size distribution data. However, higher $\rho_{eff}$ than 2.0 g cm$^{-3}$ was not considered to be realistic based on previous studies.**

|  | Helsinki: No episode | Helsinki: Inversion | Helsinki: LRT | Prague: All |
|---|---|---|---|---|
| PN (1 cm$^{-3}$) | 7 700 | 16 200 | 9 700 | 5 700 |
| PM$_{2.5}$ (µg m$^{-3}$) | 3.4 | 9.9 | 15.4 | 20.2 |
| NO (µg m$^{-3}$) | 16.1 | 29.8 | 22.2 | 28.8 |
| BC (µg m$^{-3}$) | 0.58 | 1.11 | 1.01 | 0.59 |
| $\rho_{eff}$ (g cm$^{-3}$) | 1.1 | 1.3 | 1.7 | 2.0* |

The estimated particle effective densities got higher as the contribution of regional aerosol and PM$_{2.5}$ concentration increased.
In Helsinki, $\rho_{eff}$ from local sources (without the episodes) was estimated to be close to the standard. This estimation is supported e.g., by Virtanen et al. (2006), Rissler et al. (2014), Wu et al. (2023), who reported effective densities of ultrafine particles of 1.0 (nucleation mode particles), 0.66–1.50 g cm$^{-3}$ (for 75–100 nm particles) and 0.80–0.89 in road traffic sites in Helsinki, Copenhagen and Taipei, respectively. Also, the increased $\rho_{eff}$ due to regional aerosol (i.e., increased PM$_{2.5}$) is supported by various studies which have reported effective densities of roughly 1.3–2.0 g cm$^{-3}$, average being around 1.5–1.7 g cm$^{-3}$, for

ambient particles in the accumulation mode size ranges (e.g., Virtanen et al. 2006, Levy et al. 2013, Yin et al. 2015, Lu et al. 2024). Interestingly, in Prague, according to the ELPI+ and SMPS data, the estimated $\rho_{\text{eff}}$ could have been even higher than the chosen 2.0 g cm$^{-3}$ (Fig. S5-6). However, the reported effective densities have rarely been higher than 2.0 g cm$^{-3}$ in previous studies: slightly higher effective densities have been measured mainly related to dust episodes or railway emissions (Chu and Olofsson 2018, Lu et al. 2024). On the other hand, measurements during springtime near a tram line and train station suggest

that both dust and railway emissions could have contributed to the measured aerosol in Prague. Still, the main source of particle surface area was the regional aerosol. Therefore, 2.0 g cm$^{-3}$ was considered to be the most realistic estimation for the average effective density in Prague. With this estimation, the surface area size distributions of the ELPI+ and SMPS did not match perfectly (Fig. S5) but this difference is considered to be related to the varying operating principles of the devices and measurement uncertainties. For instance, the measurement of electric current (ELPI+) versus particle number (SMPS) can lead

to differences especially with fractal-structured larger particles as discussed in Section 2.1.1 and later in Section 3.2.2. Also, the SMPS upper size limit of 500 nm may have decreased the detection efficiency of 400–500 nm particles.

### 3.2 LDSA[al] measurement method comparison

### 3.2.1 With general assumptions

The measured average LDSA[al] size distributions and concentrations with general assumptions, i.e., without corrections for the

particle effective density and hygroscopicity, are collected in Fig. 1. As can be seen, the mean particle sizes of the LDSA[al] size distributions were different which can be explained with the varying aerodynamic and mobility equivalent diameters. Still, in Helsinki, the shapes of the distributions were rather similar with each other whereas, in Prague, the ELPI+ and SMPS size distributions varied significantly, especially with the accumulation mode particles. In Helsinki, LDSA[al] concentrations measured with the DMPS and Partector were 69–74 % and 76–91 % of the ones measured with the ELPI+, respectively. In

Prague, LDSA[al] concentrations with the SMPS and Partector were 54 % and 67 % of the ones measured with the ELPI+. Scatter plots of hourly averaged data are provided in Supplementary (Fig. S16-18).

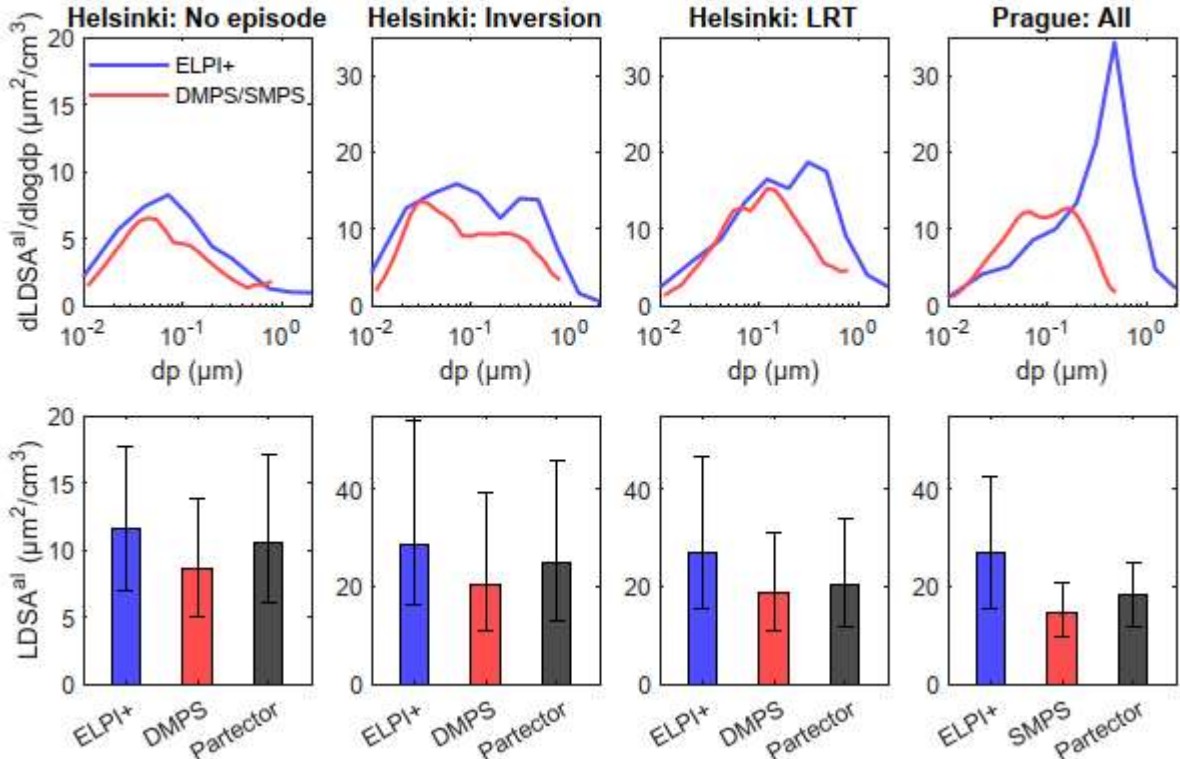

**Figure 1: Average LDSA$^{al}$ size distributions and concentrations measured with the different methods during the studied periods in Helsinki and Prague without corrections for the particle effective density nor the hygroscopic growth. The whiskers indicate the 25$^{th}$ and 75$^{th}$ percentiles of the measured concentrations. Figure LDSA$^{al}$ data collected in Table S2 (histograms Fig. S12-15). Note different y-axis range for Helsinki: No episode.**

Overall, the results in Fig. 1 show that there can be significant differences in the measured LDSA$^{al}$ concentrations with different methods even in similar kinds of urban environments if the general assumptions are applied with the data. The measured size distributions suggest that this uncertainty is especially related to the estimation of particle effective density as the differences with the size distribution methods increased as the $\rho_{eff}$ differed from the standard of 1.0 g cm$^{-3}$. The result in Prague shows that $\rho_{eff}$ does not only affect the mean size of the size distributions as it can also considerably affect the estimated absolute LDSA$^{al}$ concentration. With the Partector, the difference compared to the ELPI+ was also the highest in Prague which is likely related to the suitable size range of the measurement (20–400 nm) as at least the result with the ELPI+ suggests considerable contribution by particles larger than 400 nm on LDSA$^{al}$. On the other hand, the result in Prague also suggests that the ELPI+ may have overestimated the contribution of particles larger than 400 nm, at least if compared to the SMPS. The Partector and ELPI+ seemed to agree with each other in terms of the LDSA$^{al}$ concentration when the accumulation mode of particles did not dominate the distribution (in Helsinki) whereas the DMPS and SMPS systematically measured lower concentrations than either ELPI+ or Partector. This difference of DMPS and SMPS compared to the electrical methods is likely related to fractal structure

of particles (see section 2.1.1). Also, the narrower measurement size ranges with the DMPS and SMPS may also explain some

of the differences compared to the ELPI+ even though the measured size distributions suggest that only a small fraction of the

larger particles were undetected with the DMPS or SMPS.

### 3.2.2 Effective density correction

The $\rho_{\text{eff}}$-corrected average LDSA$^{\text{al}}$ size distributions and concentrations are collected in Fig. 2. It should be noted that the $\rho_{\text{eff}}$-correction was done only for the size distribution methods as it is not possible to correct the Partector data. In Fig. 2, it can be

observed that the differences with both size distributions and absolute concentrations decreased after the $\rho_{\text{eff}}$-correction which

supports the idea that the differences in Fig. 1 were considerably related to the standard effective density approximation.

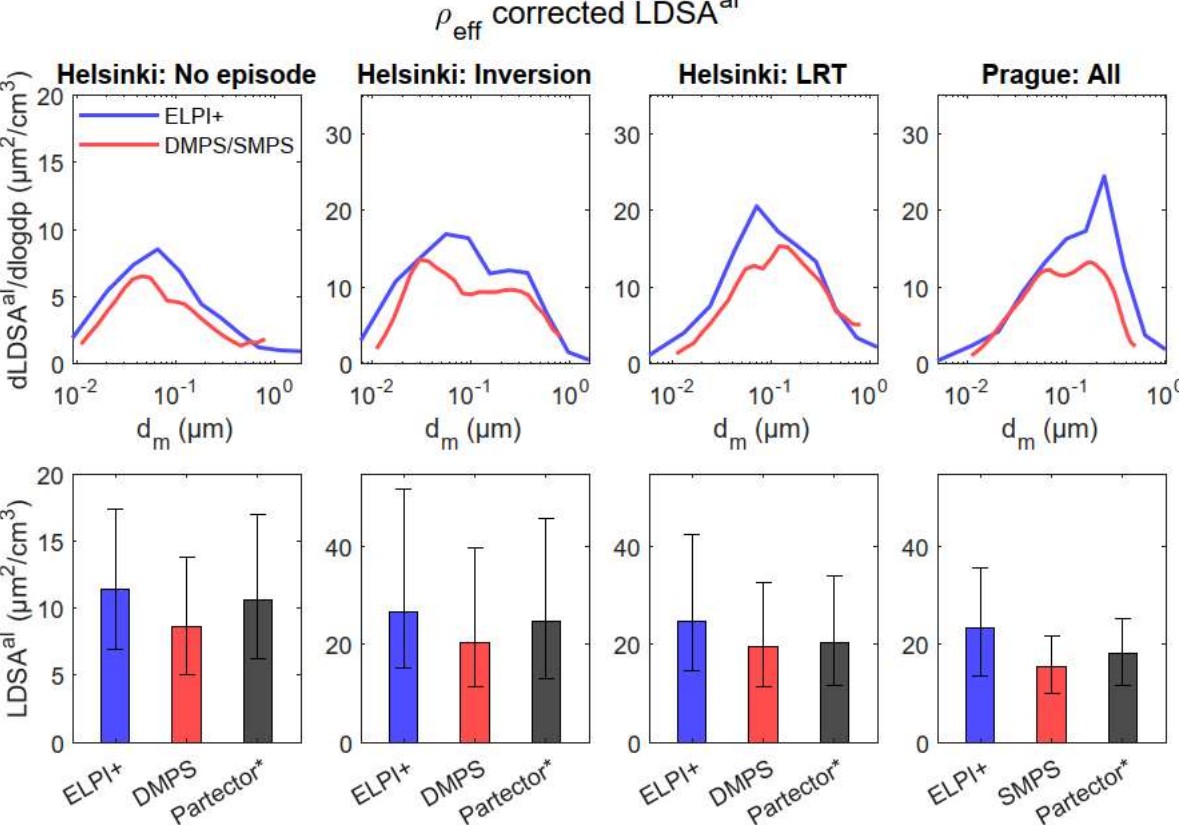

**Figure 2: Average LDSA$^{\text{al}}$ size distributions and concentrations measured with the different methods during the studied periods with corrections for the estimated particle effective density (1.1, 1.3, 1.7 and 2.0 g cm$^{-3}$, respectively). The whiskers indicate the 25$^{\text{th}}$**
**and 75$^{\text{th}}$ percentiles of the measured concentrations. *Note that the particle effective density cannot be considered with the Partector. Figure LDSA$^{\text{al}}$ data collected in Table S2 (histograms Fig. S12-15). Note different y-axis range for Helsinki: No episode.**

The average LDSA$^{\text{al}}$ concentrations with the general assumptions compared to the $\rho_{\text{eff}}$-corrected ones with the size distribution methods are collected in Fig. 3. With the ELPI+, the standard $\rho_{\text{eff}}$ assumption led to 16 % higher LDSA$^{\text{al}}$ concentration compared to the $\rho_{\text{eff}}$-corrected one in Prague, whereas, in Helsinki, the difference was less than 10 %. With the DMPS and

SMPS, the standard $\rho_{eff}$-assumption led to underestimated LDSA$^{al}$ compared to the $\rho_{eff}$-corrected one, even though the difference was 5 % or less with all the cases. The differences in the average LDSA$^{al}$ concentrations between the methods with both general assumptions and $\rho_{eff}$-correction are shown in Fig. 4. After the $\rho_{eff}$-correction, LDSA$^{al}$ with the DMPS and SMPS was 66–79 % of the ones measured with the ELPI+. The difference between Partector and ELPI+ still increased as the contribution of accumulation mode increased, similarly as with the general assumption, but the difference was less than 23 %

in all the environments (less than 18 % in Helsinki). Also, the differences in the scatter plot analysis with the hourly averaged data mainly decreased after the correction (Fig. S16-18). Thus, the $\rho_{eff}$-correction clearly decreased the differences between the methods, but it did not correct all the differences, and especially the absolute measured LDSA$^{al}$ concentrations can still be considerably different.

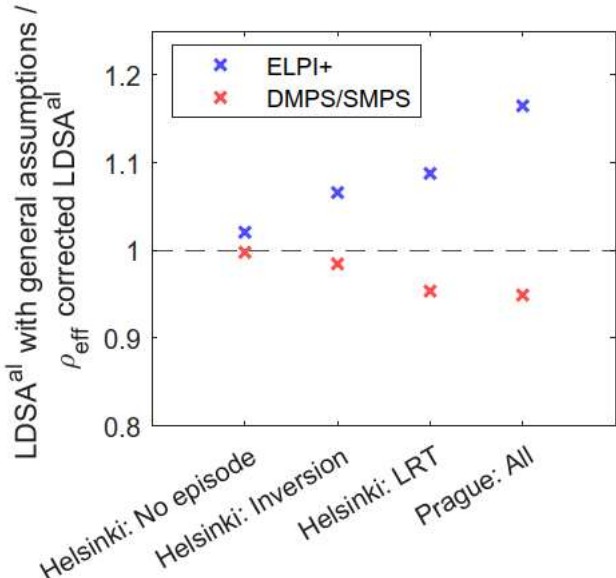

**Figure 3: Ratio between the measured LDSA$^{al}$ concentrations with and without correction for the particle effective density with the ELPI+, DMPS (Helsinki) or SMPS (Prague). See Table S2 for the measured average LDSA$^{al}$ concentrations (histograms Fig. S12-15).**

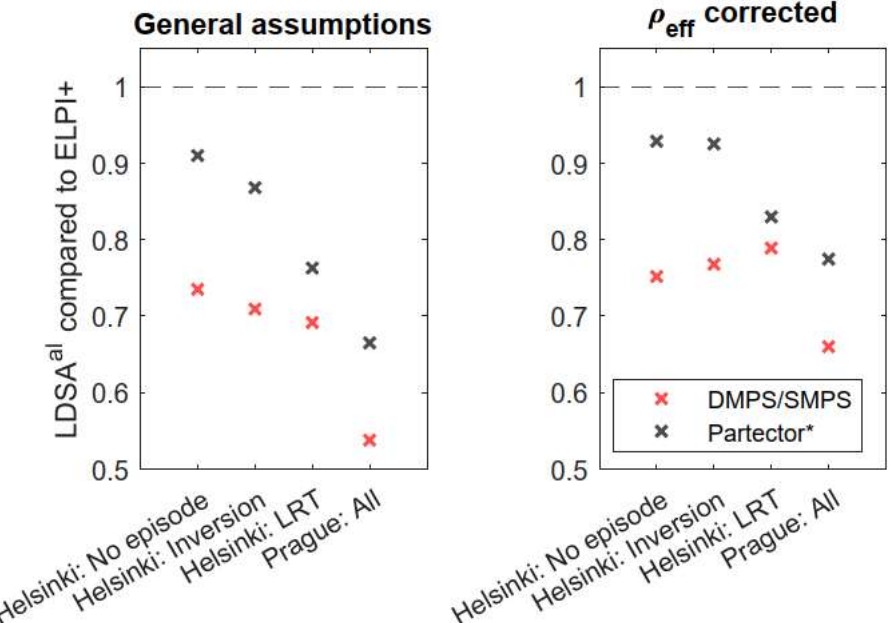

**Figure 4: Differences in the LDSA[al] concentrations measured with the DMPS (Helsinki), SMPS (Prague) and Partector compared to ELPI+ with and without correction for the particle effective density. *Note that the particle effective density cannot be considered in the Partector results. See Table S2 for the measured average LDSA[al] concentrations (histograms Fig. S12-15).**

In terms of the devices' operation principles, the results indicate that the ELPI+ is the most vulnerable to errors related to the wrongly assumed effective density in ambient conditions. This result can be explained in terms of the LDSA[al] size distributions with the size classification method of the ELPI+ which is dependent on the aerodynamic size which is the key parameter only for particles roughly larger than 500 nm in the particle lung deposition. The overestimation of total LDSA[al] concentration of the ELPI+ with the standard $\rho_{eff}$ assumption (Fig. 3) can be explained with the conversion from electric current into to LDSA[al], as the calculation considers the particles to have larger mobility equivalent size than they have in reality, causing the conversion factors into to LDSA[al] to be too high (see Lepistö et al. 2020). As seen, the majority of LDSA[al] concentration in the studied sites was attributable to particles smaller than 500 nm (mobility equivalent diameter). The DMPS, SMPS and Partector are less vulnerable to errors related to the effective density if the concentration of particles larger than 500 nm is not high as both the measurement (charging efficiency and size classification) and lung deposition efficiency are dependent on the mobility equivalent size. The slight underestimation in Fig. 3, is related to the concentrations of particles larger than about 500 nm, where the dominant deposition method changes from diffusion to impaction, causing the DMPS and SMPS to underestimate the deposition efficiency.

But, as mentioned, the $\rho_{eff}$-correction did not fix all the limitations with the measurement methods. Still, the limited effective size range of 20–400 nm with the Partector can cause considerable uncertainty with ambient aerosol, especially in regions with high PM$_{2.5}$. Also, the DMPS and SMPS seemed to underestimate the absolute LDSA[al] concentration by roughly 5–25 % after the $\rho_{eff}$-correction as well compared to the electrical methods (Fig. 4). This result agrees with a study by Chang et al. (2022)

where LDSA$^{al}$ measurements of a NSAM and SMPS were compared in Taipei. On the other hand, Chen et al. (2023) reported over 50 % differences between NSAM and SMPS in Taipei. The lower concentrations of DMPS and SMPS compared to electric methods can be explained with the measurement principles as the electric current after diffusion charger is dependent on the particle shape whereas with the DMPS and SMPS only the number of particles (assumed spherical) is measured. Thus, agglomerated structures, especially with larger particles, can cause variation with the methods (Section 2.1.1). It's worth noting

that even though the effective density can be taken into account with the size distribution methods, the size- and time-dependence of $\rho_{\text{eff}}$ is practically challenging to consider especially in typical monitoring measurements, similarly as in this study. Thus, even after an effective density correction, some instrument-dependent uncertainties related to the effective density remain in the measurement, which should be recognised when reporting LDSA$^{al}$ results. On the other hand, with DMPS and SMPS, the approximation of one averaged effective density for all the particles does not cause considerable uncertainties in

the results due to the fact that both measurement method and lung deposition are mainly dependent on the mobility equivalent size of particles. This is demonstrated in Fig. S19, where example comparisons of DMPS and SMPS data with averaged, standard, and size-dependent effective densities in Helsinki and Prague are shown. However, with the ELPI+, the operation principle does not fundamentally enable utilisation of size-dependent effective density (see 2.3), which should be acknowledged. Still, this uncertainty of ELPI+ can be estimated by comparing the density-corrected results to the DMPS/SMPS

results which are not as vulnerable to errors in terms of varying effective density.

### 3.2.3 Effect of particle hygroscopicity

In Fig. 5, the average LDSA$^{al}$ size distributions with corrections for both particle effective density and hygroscopic growth are compared to the ones without any corrections and with correction only for the effective density. As seen, the hygroscopicity had a strong effect on LDSA$^{al}$ size distributions especially with particles larger than 400 nm. In general, hygroscopicity

correction decreased the lung deposition efficiency of particles smaller than 200 nm whereas it increased the deposition efficiency of particles larger than 200–400 nm (see also Fig. S7). Quite surprisingly, with the ELPI+, $\rho_{\text{eff}}$- and hygroscopicity-corrected LDSA$^{al}$ size distributions were rather close to the ones with the general assumptions. This result can be explained by the fact that the estimated LDSA$^{al}$ of particles larger than 400 nm decreased after the $\rho_{\text{eff}}$-correction whereas the hygroscopicity correction increased the estimated lung deposition of these larger particles. With the SMPS and DMPS, this

similar behaviour did not occur as the $\rho_{\text{eff}}$-correction did not dramatically change the measured size distribution. Hence, with the SMPS and DMPS, the hygroscopicity corrected size distributions varied more compared to the ones with the general assumptions. In general, after the hygroscopicity-correction, the shapes of the size distributions with both methods agreed reasonably well with each other.

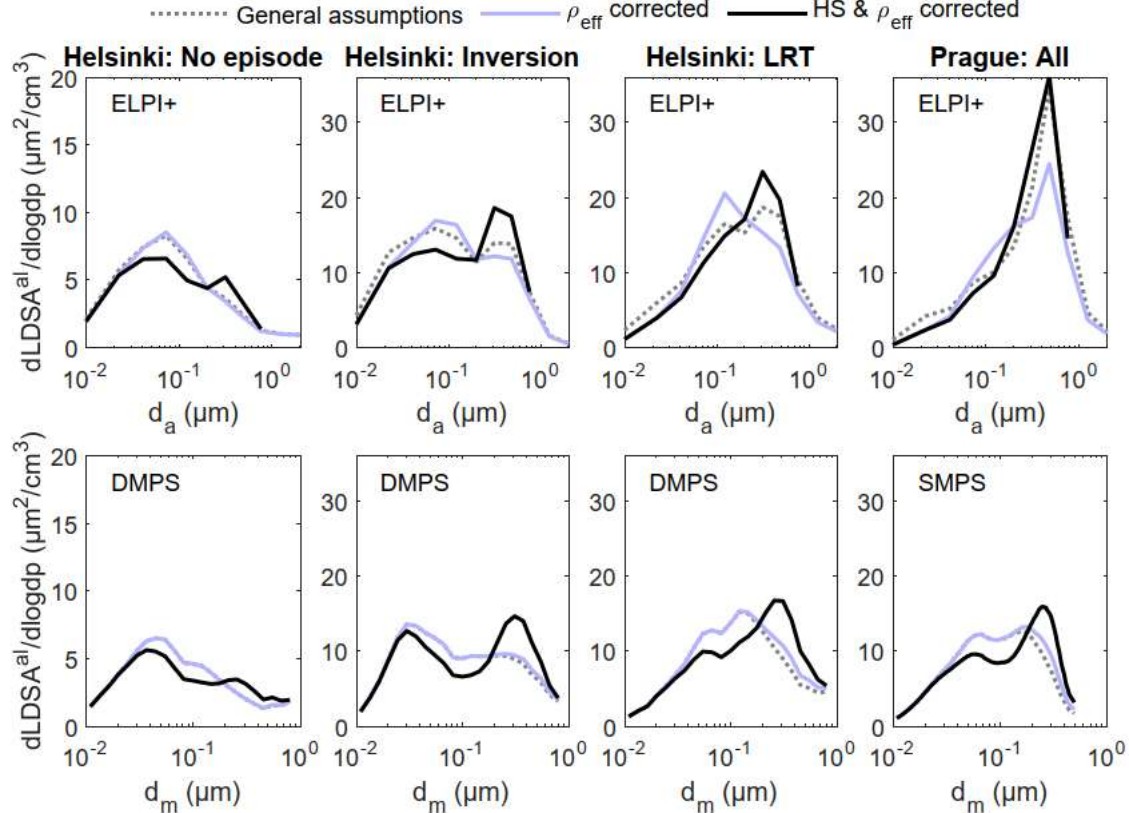

**Figure 5: Measured average LDSA[al] size distributions without corrections for the particle effective density and hygroscopic growth (from Fig. 1), with correction for the effective density (from Fig. 2), and with corrections for both effective density and hygroscopic growth.**

Even though the hygroscopicity-correction can considerably change the estimated LDSA[al] size distributions, the effect on the measured absolute LDSA[al] concentration was less significant which can be seen in Fig. 6. Note that the correction was not done for the Partector data. With the ELPI+, LDSA[al] concentration with the general assumptions was 107–114 % of the hygroscopicity-corrected result in all the cases. With the DMPS and SMPS, LDSA[al] with general assumptions was 95–104 % of the ones with the hygroscopicity-correction. This result can be explained due to the balancing effects of particles smaller than 200 nm and larger than 200–400 nm in terms of the hygroscopicity correction. Thus, by a coincidence, accuracy of the absolute LDSA[al] concentration measurement was not significantly affected due to the particle hygroscopicity, which can also be seen in the hourly averaged scatter plots (Fig S16-18). However, it's worth noting that this result may depend on the location and urban environment. For example, high concentration of accumulation mode particles can potentially cause underestimation of LDSA[al] without hygroscopicity correction. Also, it's important to note that the hygroscopicity correction still affected the relationship between the studied instruments (Fig. 6). In Helsinki, the DMPS and Partector measured 78–83 % and 87–103 % of the LDSA[al] measured with the ELPI+ after the hygroscopicity-corrections, respectively. However, in Prague, the ratios

dropped to 61 % and 73 % compared to the ELPI+, respectively. Thus, the uncertainty related to the particle hygroscopicity increased as the concentrations of the accumulation mode particles increased, similarly as with the particle effective density.

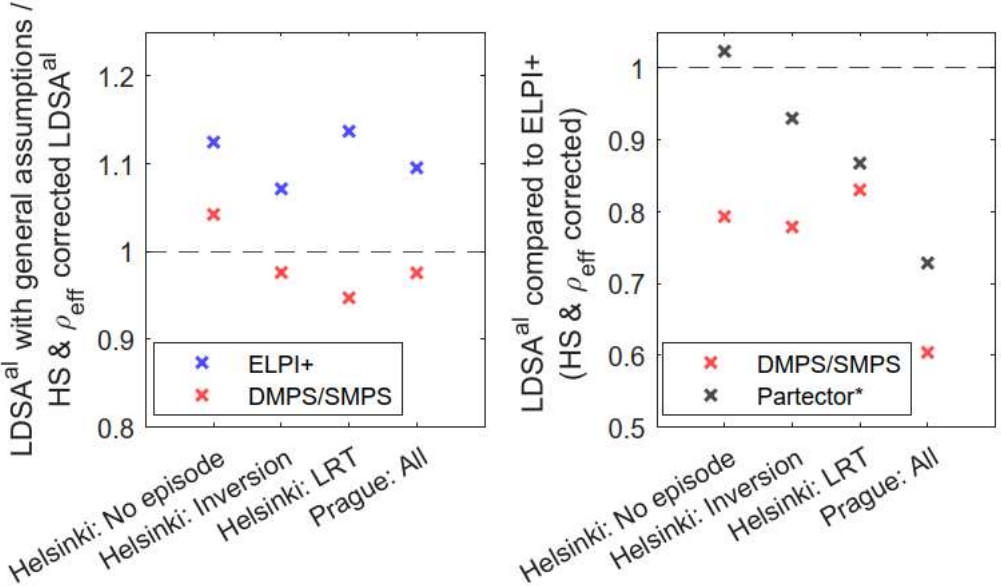

**Figure 6: a) Ratio between the measured LDSA<sup>al</sup> concentrations with and without corrections for the particle effective density and hygroscopic growth. b) Differences in the LDSA<sup>al</sup> concentrations measured with DMPS, SMPS and Partector compared to ELPI+ with corrections for the particle effective density and hygroscopicity. *Note that the corrections cannot be done for the Partector results. See Table S2 for the measured average LDSA<sup>al</sup> concentrations (histograms Fig. S12-15).**

### 3.2.4 Location-dependence with the electrical particle sensors

As the electrical particle sensors are at least the most affordable solution to monitor LDSA<sup>al</sup> concentrations, it should be noted that in addition to the effective density and hygroscopicity, varying particles sizes also affect the performance of the sensors, as can be seen also with the Prague data where the contribution of particles larger than 400 nm were underestimated with the Partector. However, varying particle size distributions within the suitable 20–400 nm size range can also affect the accuracy of the measurement as the response between the diffusion charged current and LDSA<sup>al</sup> is not linear (e.g., Todea et al. 2015), and the particle size distributions can considerably vary depending on the nearby emission sources (e.g., Masiol et al. 2017, Harni et al. 2022, Lepistö et al. 2023). In Fig. 7, the average LDSA<sup>al</sup> concentrations with the ELPI+ and Partector from the measurements in Helsinki and Prague as well as in Tampere and Düsseldorf are compared. The data in Fig. 7 were not corrected based on the effective density and the hygroscopicity as data were not available for these corrections in Tampere and Düsseldorf measurements.

In Fig. 7, it can be seen that the devices agreed rather well with each other in locations with high PN concentrations whereas the difference increased in locations with low PN and high $PM_{2.5}$. The difference with high $PM_{2.5}$ can be explained with the contribution of accumulation mode particles larger than 400 nm which typically increase as a function of $PM_{2.5}$, and, hence, the Partector underestimates the absolute LDSA<sup>al</sup> concentration. On the other hand, the ELPI+ may overestimate the

contribution of accumulation mode particles without correction for the effective density as seen in Fig. 1–3, but this overestimation alone does not explain the whole differences in Fig. 7. The high PN concentration indicates contribution of

ultrafine particles, which can be efficiently measured with the Partector, explaining why the differences decreased with higher PN concentrations. It is also worth noting that the performance of the Partector was good (> 80 % of ELPI+) in all the studied locations with high PN, including road traffic sites, airport and effects of residential wood combustion (see Table S3). Therefore, it seems that the varying particle sizes of the nearby emission sources does not dramatically affect the accuracy of the sensor measurement. For example, in the airport, LDSA$^{al}$ can be significantly contributed to by < 50 nm particles whereas,

in road traffic sites, the peak of the LDSA$^{al}$ size distribution is around 100 nm (Lepistö et al. 2023). On the other hand, in all the studied sites, the Partector measured lower concentrations than the ELPI+ (also after density corrections in Helsinki and Prague), suggesting systematic difference between the methods, which may e.g., be related to different lung deposition models in the device calibrations. Despite the underestimation of the Partector in sites with high PM$_{2.5}$ and low PN, the correlation of the measured concentrations with the ELPI+ and Partector was still strong in all the studied sites (R$^2$: 0.85-0.98, Fig. S20-22),

showing that the challenges are mainly related to the utilised calibration factor.

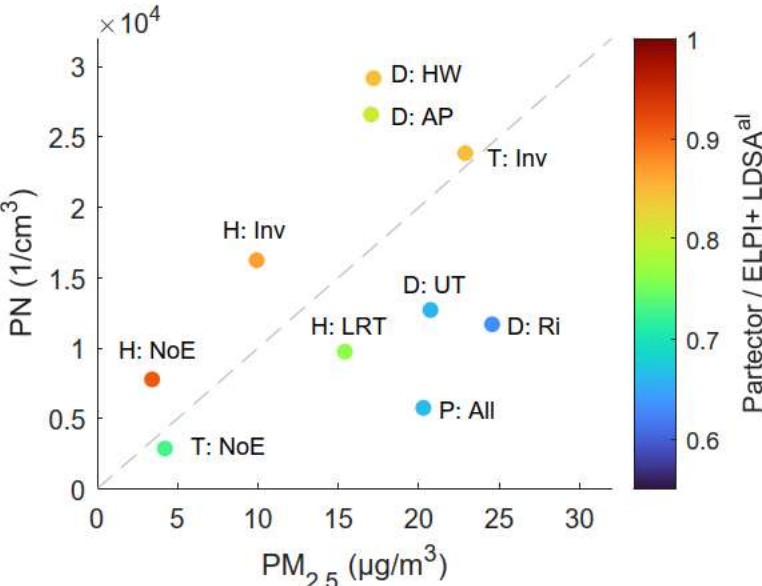

**Figure 7: Comparison of LDSA$^{al}$ concentrations measured with the Partector and ELPI+ as a function of particle number (PN) and PM$_{2.5}$ concentrations, which were calculated from the ELPI+ data. Each dot represents individual measurements in different locations (see Table S3). D: AP (Airport), D: HW (Highway), D: UT (Urban traffic) and D:Ri (River) indicate measurements in**

**Düsseldorf. H: NoE (No episodes), H: Inv (Inversion), H: LRT and P: All indicate the measurements in Helsinki (H) and Prague (P). T: NoE and T: Inv indicate the measurements in Tampere.**

The results in Fig. 7 show that the comparison of only electrical particle measurements of LDSA$^{al}$ can still be complicated due to the differences in the local and regional pollution levels. Therefore, comparison of LDSA$^{al}$ measurements even with the

same device can be challenging in different locations depending on the regional pollution levels. On the other hand, it is worth

discussing whether the LDSA[al] sensor measurement should be considered to represent only the LDSA[al] attributable to particles smaller than 400 nm which would reduce this uncertainty. In this scenario, however, the particles larger than 400 nm should be removed from the sample, or the contribution of these particles should be estimated e.g., by measuring the regional background concentration far away from any pollution sources. In Fig. S23, similar comparison as in Fig. 7 between the ELPI+

and Partector was done by considering only particles smaller than 400 nm with the ELPI+. As a result, the Partector reported 5–29 % higher LDSA[al] concentrations than the ELPI+ which shows that utilisation of the sensors only as an indication of LDSA[al] concentration attributable to particles smaller than 400 nm is problematic if the contribution of larger particles is not considered.

### 3.2.5 Summary of the comparisons

As a summary, comparison of LDSA[al] results with different measurement methods can be complicated (see also Fig. S24). From a technological point-of-view, especially the particle size and effective density are important parameters when comparing the different methods. When considering the particle size, high concentrations of particles larger than 400 nm cause significant underestimation of absolute LDSA[al] concentration with the electrical particle sensors. The size distribution methods can consider varying particle sizes but, on the other hand, the measurement size ranges can be different (similarly as in this study)

which complicates the comparison. In this study, the uncertainty related to varying measurement size ranges seemed, however, to be minimal. In addition, with the size distribution methods, especially the ELPI+ was vulnerable to errors related to the wrongly assumed particle effective density: the ELPI+ overestimated the LDSA[al] concentration roughly up to 20 %, whereas, with the DMPS or SMPS, the uncertainty related to effective density was less than 5 %, which is likely the case with the electrical particle sensors as well. However, when considering the conversion from the measured quantity to LDSA[al], the

DMPS and SMPS seemed to systematically underestimate the absolute LDSA[al] concentration by roughly 5–25 % compared to the electrical methods which likely better represent the actual surface area of particles (see also Chang et al. 2022, Chen et al. 2023). Thus, all the studied methods have both strengths and disadvantages in LDSA[al] measurement, and it is not possible to justifiably claim any of the methods to be the best method for LDSA[al] measurement in general. Therefore, the disadvantages of each method should be carefully considered when reporting LDSA[al] results.

In addition, the effect of particle hygroscopicity should be recognised when reporting LDSA[al] results. Especially with the LDSA[al] size distributions, hygroscopic growth of particles can significantly change the result in ambient conditions. On the other hand, in terms of the absolute concentrations, neglected hygroscopicity did not considerably change the results due to the balancing effects with different particle sizes. Also, the measured LDSA[al] size distributions with the ELPI+ were surprisingly close without any corrections and with corrections for both the effective density and hygroscopicity as the effective

density and the hygroscopicity had balancing effects in the result. With the DMPS or SMPS this similar phenomenon did not occur.

When considering the suitability of LDSA[al] in air quality monitoring measurements, it should be noted that the particle effective density and especially the hygroscopicity are practically challenging to consider. Thus, in comparison with other

commonly utilised metrics (like PM$_{2.5}$ or PN), there are considerably higher uncertainty with LDSA[al] even if the measurements have been conducted with the same device. On the other hand, the challenges related to LDSA[al] measurement seemed to become more relevant with the larger particles. For instance, the effective density of particles emitted from nearby local pollution sources is rather close to the standard. Also, the ultrafine or soot particles are typically hydrophobic, and the hygroscopic growth rates start to increase considerably with particles larger than 200–400 nm (Vu et al. 2015). In addition, the result in Fig. 7 and Table S3 show that electrical particle sensors are accurate in various urban environments despite different particle size distributions as long as the particles are mainly smaller than 400 nm. In terms of particle health effects, the relevance of surface area is likely the highest with the smaller ultrafine and soot particles (Oberdorster 2005, Schmid and Stoger 2016, Hakkarainen et al. 2022) whereas with, larger particles, secondary aerosol and soluble particles, the health effects have been strongly associated with the mass concentration (e.g., Lakey et al. 2016, Lin et al. 2016, Yu et al. 2022, Yang et al. 2023). Therefore, it is uncertain whether surface area deposition is relevant in terms of larger and soluble particles. Thus, in terms of monitoring the effects of nearby local pollution sources in a dense air quality monitoring network, LDSA[al] should be well a suitable and potential metric in terms of the particle health effects. This idea is supported by Fig. S24-25, where the different methods agreed reasonably well in terms of LDSA[al] attributable to particles smaller than 400 nm, and the effective density or hygroscopicity corrections did not considerably change the result. Still, it should be acknowledged that especially with the electrical sensors, particles larger than 400 nm are still measured and, therefore, can affect the accuracy of the measurement. Thus, with the electrical sensors, it would be reasonable to remove the larger particles from the sample or to utilise regional background measurements of LDSA[al] to reduce the uncertainty related to larger accumulation mode particles in the result.

## 4 Strengths and limitations

The main strength of this study is that it provides comprehensive information of the differences between different LDSA[al] measurement methods in ambient measurements which have not been typically considered in previous studies. Therefore, the results help the interpretation of previous and future LDSA[al] studies conducted with different instrumentation. However, the uncertainties related to the analysis of this study, e.g., related to the determination of the particle effective density and hygroscopic growth, should be acknowledged. In this study, it was possible to estimate the average effective density of particles by comparing the ELPI+ and DMPS/SMPS size distributions as well as the effects of hygroscopicity based on a review by Vu et al. (2015). However, these parameters have spatiotemporal variability, and they depend on the particle size and composition. In general, these factors are challenging to determine (like $\rho_{eff}$ in Prague), especially when considering the typical air quality monitoring measurements. Hence, not all the effects of particle effective density nor hygroscopicity were recognised in the analysis, and thus the results of these parameters should be considered to be indicative. On the other hand, with the DMPS and SMPS, the uncertainty due to average density assumption is less significant (Fig. S19), and the uncertainty of ELPI+ can be estimated by comparing the results to the DMPS and SMPS size distributions. Also, the analysis agree with or are based on

existing literature, and, therefore, the analysis can be considered to be reasonable. It is important to note that, according to the authors' knowledge, the effects of these parameters have not been previously analysed in terms of ambient LDSA[al] measurements. Hence, the results provide valuable information of these effects on the different LDSA[al] measurement methods in ambient conditions even though the analysis includes necessary approximations. The same principle applies to the results

of this study also in general: all the studied instruments had both strengths and weaknesses, and, hence, it is not possible to justifiably claim any of the methods to be the best in terms of LDSA[al] measurement. Still, the results clearly show how the devices' operation principles or the varying particle characteristics can affect the reported results in varying ambient conditions, which is crucial when comparing the results of different studies.

Additionally, differences with the utilised particle lung deposition function in the LDSA[al] measurement should be

acknowledged. The particle lung deposition can be estimated with different models, and, e.g., both the ICRP and the multiple-path dosimetry model (MPPD, Asgharian et al. 2001) models have been both frequently utilised in LDSA[al]-studies (e.g., Chang et al. 2022, Teinilä et al. 2022, Liu et al. 2023, Chen et al. 2023). In addition, the chosen input parameters for the models, like the human anatomy and physical activity, affect the lung deposition estimations. Thus, different LDSA[al] measurement methods can have differences in the applied lung deposition functions. This uncertainty is also difficult to estimate as, e.g., the Partector

and ELPI+ report LDSA[al] based on their own calibration (Fierz et al. 2014; Lepistö et al. 2020), whereas with the SMPS and DMPS, the utilised model is chosen by the user (which can be done with the ELPI+ as well). On the other hand, the different models agree reasonably well with each other, especially in terms of the shape of the deposition curve (e.g., Hofmann et al. 2011). However, for example, in Fig. 7, Partector systematically measured lower concentrations than ELPI+ in all the studied sites, which suggest that varying deposition models in the devices' calibration could also have had an effect on the results.

Still, the uncertainties related to the deposition models are likely considerably less significant than the studied effects of particle effective density and hygroscopicity. Still, it would be beneficial to determine common practices for LDSA[al] measurement in general regarding the utilised lung deposition function. Also, further studies on the effects of the chosen lung deposition model, and its parameters, in terms of LDSA[al] measurement with different devices would be beneficial to better understand how well the current measurement methods represent actual lung deposition, in addition to the effects of effective density and

hygroscopic growth.

In all, the results indicate that utilisation of LDSA[al] as a monitored metric in air quality monitoring measurements is complicated but also holds potential. The results suggests that the main challenges of the measurement start to have a considerable effect on the results only with high concentrations of accumulation mode particles larger than 200–400 nm. Also, the relevance of surface area in terms of the adverse health effects is not as evident anymore with larger and soluble particles

compared to solid ultrafine particles or soot. Therefore, LDSA[al] could be a suitable parameter for detecting the spatial differences in the particulate pollution within cities as the effects of nearby pollution sources, like traffic, are commonly observed with ultrafine and soot particles that are smaller than 200 nm. As the current scientific evidence highlights the need for dense air quality monitoring networks and implementation of new parameters like PN and BC in the monitoring, the sensitive and reasonably accurate measurement of ultrafine and soot particles with LDSA[al] could provide a cost-efficient

method for monitoring measurements, e.g., with the electrical particle sensors. The results of this study suggest that there should not be significant dependence of the urban environment in terms of the performance of the electrical particle sensors as long as the local pollution dominates the sample, and the effects of the larger accumulation mode particles are taken into account in the analysis. In this study, the detailed comparison was, however, only done for road traffic environments in lowly or moderately polluted regions. Also, the measurements of this study were rather short, and conducted only during certain seasons. Thus, there is still a need for studies of particle effective density, hygroscopic growth, and particle size distributions along with LDSA[al] measurement in different urban environments and in highly polluted regions, including long-term data, to better understand the universal suitability and behaviour of the metric.

## 5 Conclusions

The results of this study show that comparison of ambient LDSA[al] measurements with different instruments and in different locations can be complicated. The comparisons of this study included one electrical particle sensor (Partector) and two different size distribution approaches (ELPI+ and DMPS/SMPS). Especially, the particle size, effective density and hygroscopicity can considerably affect the LDSA[al] measurement, and the effects are not the same with different devices. On the other hand, when considering all the required parameters for the measurement, e.g., the particle effective density, the differences between the methods decreased considerably, but not completely. However, the challenges of the measurement were mainly related to the accumulation mode particles larger than 200–400 nm, for which surface area may not be as relevant in terms of the adverse health effects as with smaller ultrafine particles or soot. Therefore, regardless of the method, LDSA[al] should be well suitable when considering its utilisation to detect the effects of nearby local pollution sources in dense air quality monitoring networks as long as the effects of larger particles are addressed either by removing them from the sample or also measuring the regional background concentration. Still, further research on the relevance of surface area in terms of larger and soluble particles as well as determination of common practises for LDSA[al] measurement related to the utilised deposition model and its parameters are needed to better understand the relevance and improve the suitability of LDSA[al] in terms of air quality monitoring.

**Author contribution (Credit)**

Teemu Lepistö: Conceptualization, Methodology, Validation, Formal Analysis, Investigation, Data Curation, Writing - Original Draft, Writing - Review & Editing, Visualization

Henna Lintusaari: Methodology, Investigation, Data Curation, Writing - Review & Editing

Laura Salo: Investigation, Data Curation, Writing - Review & Editing

Ville Silvonen: Investigation, Data Curation, Writing - Review & Editing

Luis M.F. Barreira: Investigation, Writing - Review & Editing

Jussi Hoivala: Investigation, Writing - Review & Editing

Lassi Markkula: Investigation, Writing - Review & Editing

Jarkko V. Niemi: Resources, Data Curation, Writing - Review & Editing

Jakub Ondracek: Resources, Data Curation, Writing - Review & Editing

Kimmo Teinilä: Methodology, Writing - Review & Editing

Hanna E. Manninen: Resources, Writing - Review & Editing

Sanna Saarikoski: Methodology, Investigation, Writing - Review & Editing, Project administration, Funding acquisition

Hilkka Timonen: Methodology, Writing - Review & Editing, Project administration, Funding acquisition

Miikka Dal Maso: Writing - Review & Editing, Project administration, Funding acquisition

Topi Rönkkö: Methodology, Writing - Review & Editing, Project administration, Funding acquisition

## Competing interests

Some authors are members of the editorial board of journal Aerosol Research.

## Acknowledgements

This study is part of the AEROSURF-project, funded by the Research Council of Finland (grant no. 356752).

This work has received funding from the European Union's Horizon 2020 research and innovation programme under grant

agreement No 814978 (TUBE: Transport-derived ultrafines and the brain effects).

The study has received funding from the BC Footprint (530/31/2019) and Future Spaces (33250/31/2020) -projects, funded by

Business Finland, participating companies, and municipal actors.

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
