# Peer review of "Comparison of size distribution and electrical particle sensor measurement methods for particle lung deposited surface area (LDSA[al]) in ambient measurements with varying conditions"

_Aerosol Research, 2024_

## Author Response (AR1)

**Author response to Reviewer #1**

The manuscript presents a comprehensive comparison of different methods for measuring particle lung deposited surface area (LDSA) in ambient air, focusing on the challenges and uncertainties associated with these measurements. The study provides valuable insights into the performance of various LDSA measurement techniques under different environmental conditions and particle characteristics. The authors have conducted a thorough analysis, considering factors such as particle effective density and hygroscopicity, which are often overlooked in LDSA measurements.

- We thank you for commenting our manuscript. We believe that your comments have helped us to improve the discussion and clarity of the manuscript. Our responses are provided below. We have also made changes in the revised manuscript. We hope that our responses and changes are satisfactory.

**General Comments:**

The abstract could benefit from a more detailed explanation of the challenges in estimating lung deposition of accumulation mode particles and the acceptable differences between methods when considering only ultrafine particles (UFP) and soot.

- Thank you for this suggestion. The last sentences of the abstract were modified: *"The challenges were especially related to the accumulation mode particles roughly larger than 200–400 nm **for which the dominant deposition mechanism in the lung changes from diffusion to impaction, and the particle effective density and hygroscopicity tend to increase**. On the other hand, the results suggest that the differences between the methods are reasonably low when considering only ultrafine and soot particles**, which have effective density closer to the standard (1.0 g/cm$^3$) and are more hydrophobic**, highlighting the suitability of LDSA$^{al}$ as a monitored metric when estimating spatial differences in the particulate pollution within cities."*
- Also, some parts of the abstract were slightly shortened due to the added text.

Some statements in the introduction require revision or additional context to avoid oversimplification.

- We have carefully checked the introduction. Detailed changes are mentioned later in our responses to the Specific comments.

The manuscript would benefit from more quantitative analysis to support some of the key conclusions, particularly regarding the effects of particle effective density and hygroscopicity corrections.

- We have added additional analysis regarding the size-dependence of particle effective density. Also, the discussion of the results has been modified in the revised manuscript. Please, see our responses to Specific comments.

Some technical aspects of the measurement devices and data processing methods need further clarification.

- We have added information of the devices and the methods according to the Specific comments (see our responses there).

The discussion of the impact of particle effective density and hygroscopicity on LDSA measurements needs to be supported by additional data analysis.

- As mentioned, we have modified the discussion of the results in the revised manuscript. Please, see our responses to Specific comments.

**Specific Comments:**

Line 64: The statement about PM2.5 being relatively more harmful near local pollution sources like traffic is oversimplified and should be revised or removed.

- The statement has been changed to: "***For example, it has been suggested that within-city PM$_{2.5}$ dose-response gradients are steeper than between-city gradients, emphasising the role of near-source exposure (e.g., to traffic) in terms of adverse health effects of particles (Segersson et al. 2021).***"

Line 70: The concept of LDSA should be more precisely defined, acknowledging that it can refer to different regions of the respiratory system, not just the lung alveoli. It should also be clarified that LDSA refers to a surface area concentration.

- In the revised manuscript, it is mentioned that in different studies LDSA can be referred also to other respiratory tract regions than the alveoli. It is also mentioned that LDSA refers to surface area concentration. Here, we utilise LDSA$^{al}$ notation to highlight that we refer only to alveolar LDSA (which is also the most commonly measured). The new text: ***"It's worth noting that, in different studies, LDSA can also be referred when considering other respiratory tract regions than the alveoli (e.g. Liu et al. 2023). Here, the notation LDSA$^{al}$ is used to clarify that only alveolar deposition is considered in this study (see Lepistö et al. 2023)."***

Line 133: More information about the Partector's design, particularly regarding ion trapping, would be helpful to understand potential influences on the charging current and calculated LDSA. The authors should address how the extrinsic charging efficiency, which is affected by particle losses in the charger, impacts the measured charging current and, subsequently, the calculated LDSA. This is crucial because particle losses can significantly alter the relationship between the charging current and the actual LDSA, potentially leading to inaccuracies in the final LDSA measurements.

- Description of the diffusion charging efficiency and its size dependence has been added in the revised manuscript. We agree that ion trapping is an important factor in the operation of a diffusion charger sensor. However, in a study by Fierz et al. (2014), which represents the Partector operation principle, the ion trapping of the device is not further described. However, the role of ion trapping is now shortly mentioned by citing Fissan et al. (2006) in the revised text. It's worth noting that some other particle sensors (like the new Partector 2 Pro), may alter the parameters of the charger (like the ion trap) to estimate particle sizes better, but this is not the case with the original Partector (used in

this study), and therefore, this topic is not discussed in-detail. The changes:
"*The Partector (Naneos GmbH, Fierz et al. 2014) represents the electrical particle sensor measurement method for LDSA[al]* **which is based on detecting the electric current caused by the sampled particles after a diffusion charger. The diffusion charging efficiency is determined as a multiplication of the number of elementary charges of a particle after charging (n), and the probability of a particle to penetrate through the charger (P). The product, Pn, is dependent on the particle mobility equivalent size with an exponent varying typically between 1.1– 1.9 (Dhaniyala et al. 2011, Järvinen et al. 2014). Due to lucky coincidence, the charger efficiency correlates reasonably well with LDSA[al] of a single particle in a size range roughly from 20 nm to 400 nm, which can, however, be altered slightly by adjusting the ion trap voltage of the charger (Fissan et al. 2006)**."**

Line 147: The statement about ELPI+ measurement requiring estimation of particle effective density needs clarification or correction. ELPI+ particle size distribution measurements are based on aerodynamic sizing, which inherently incorporates information about particle density. Therefore, it's not immediately clear why additional estimation of particle effective density would be required. The authors should explain this apparent discrepancy or revise their statement if it's not accurate. If there are specific reasons why effective density estimation is still necessary for LDSA calculations with ELPI+ data, these should be clearly explained.

- In ELPI+, the charger efficiency depends on the on the mobile equivalent size, but the size classification depends on the aerodynamic size. When measuring e.g., particle number, it is needed to know an average current caused by one particle collected onto the impactor stage to convert the electrical current data to particle number. As only the aerodynamic size is known, the effective density needs to be estimated to know the average current caused by a single particle, and, therefore, to convert the measured electric current into the wanted quantity accurately. In the revised manuscript:
"*As the particle charge after the diffusion charger,* **and, therefore, the measured electric current,** *is dependent on the particle mobility equivalent diameter, and the size classification is dependent on the aerodynamic diameter, the ELPI+ measurement requires estimation of the particle effective density* **to estimate the average electric current caused by a single particle collected onto a impactor stage and to convert the measured current to other quantities accurately**."**

Line 155: A brief description of the ICRP model used for the lung deposition function should be included.

- In the revised text, parameters for the ICRP-model calculations (gender and physical activity) are provide in new Table S1. Also, the ICRP-model is shortly introduced:
"**The ICRP-model is a semi-empirical regional compartment lung-deposition model which considers the human respiratory tract as a series of filters and utilises measured data with human volunteers**".

More detailed information about the conversion factor/process for the Partector is needed better to evaluate the differences between its measurements and other methods.

- Partector utilises a constant conversion factor from electric current to LDSA[al]. This factor has been determined based on the response coefficient at 100 nm size (see Fierz et al. 2014). This point is also added to the revised text:
  *"The Partector first charges the sampled particles in a diffusion charger and then converts the detected electric current caused by the sampled particles into LDSA[al] concentration with a single calibration factor. **The chosen calibration factor is the response coefficient between the electric current and LDSA[al] at 100 nm, which typically is close to the peak size of LDSA[al] size distributions in urban environments (Fierz et al. 2014)."***

Table 1: The low PN concentration and high density reported for Prague require attention/explanation and careful interpretation.

- The site micro-environment is the most likely explanation for the lower PN measured in Prague compared to Helsinki. Also, note that geometric mean is used in the results, which typically gives slightly lower values than the arithmetic mean. In the revised manuscript, we have added additional information of the Prague site (2.2.2.):
  "***In comparison with the Helsinki street canyon site, the measurement site was in an open environment in a preschool yard behind a fence, which limited the direct effects from the nearby traffic***."
  Also, in the results (3.1):
  *"In general, the contribution of the nearby road traffic was clearer in Helsinki than in Prague **due to the shorter distance from the passing vehicles to the measurement site, partly explaining the** relatively higher average PN, NO and BC concentrations compared to PM$_{2.5}$."*
- About the particle effective density, we acknowledge that the effective density determination includes necessary approximations (e.g., the averaged density for all particles). This approach and its limitations are explained in our responses to later comments. Also, the limitations of effective density estimations are addressed in the Strengths and limitations: "*In this study, it was possible to estimate the average effective density of particles by comparing the ELPI+ and DMPS/SMPS size distributions as well as the effects of hygroscopicity based on a review by Vu et al. (2015). However, these parameters have spatiotemporal variability, and they depend on the particle size and composition. In general, these factors are challenging to determine **(like ρ$_{eff}$ in Prague)**, especially when considering the typical air quality monitoring measurements. Hence, not all the effects of particle effective density nor hygroscopicity were recognised in the analysis, and thus the results of these parameters should be considered to be indicative.*"

Lines 343-346: The conclusion about uncertainty related to particle effective density estimation needs stronger support from the data presented. Several issues should be addressed:

The unusually low PN concentration in Prague needs explanation.

- Please, see the response to the previous comment.

The limitations of using an average constant effective density over a wide size range should be discussed more thoroughly.

- The use of average constant density can be justified based on the instrument operation principles. The DMPS and SMPS results could be corrected by using a size-dependent effective density, but, with the ELPI+, the operation principle does not fundamentally enable the use of size-dependent effective density in the analysis (because the cascade impactor: collection efficiencies of different stages would overlap) without extensive simulation which has not been provided by the manufacturer according to our knowledge. The use of size-dependent effective density is not possible with the device software/operation either. Thus, the constant effective density estimation is the best that can be done with the data (and this is also very common issue with all urban aerosol measurements as there are only rarely enough instrumentation in monitoring sites to measure size-dependent density). We are however interested to hear approaches of how to take varying particle density profiles into account with ELPI+ data.

- We have added discussion of this limitation in the manuscript under Section 3.2.2. We also conducted an additional sensitivity analysis (new Fig S19) for the DMPS and SMPS data by utilising size-dependent effective density (approximated according to the studies of Virtanen et al. (2006), Rissler et al. (2014), Yin et al. (2015) and Lu et al. (2024)). In Fig S19, it can be seen that the size-dependent effective density does not considerably change the DMPS/SMPS results, which can be explained because of the dependence of mobility equivalent diameter with both the measurement method and lung deposition (also discussed in the manuscript). The scaled effective density function is not based on our data and is only an approximation, but we believe that this additional analysis shows that the average effective density approach does not cause considerable uncertainties with the DMPS and SMPS. Still, with the ELPI+, this uncertainty cannot be estimated in a justifiable way. However, the ELPI+ result can be compared to the DMPS/SMPS results, which do not have major uncertainty related to effective density. Also, it needs to be mentioned that, with ELPI+, constant effective density is always required, and thus, it is also reasonable to consider ELPI+ results with an average effective density.

- Added text (3.2.2.): "*On the other hand, with DMPS and SMPS, the approximation of one averaged effective density for all the particles does not cause considerable uncertainties in the results due to the fact that both measurement method and lung deposition are mainly dependent on the mobility equivalent size of particles. This is demonstrated in Fig. S19, where example comparisons of DMPS and SMPS data with averaged, standard, and size-dependent effective densities in Helsinki and Prague are shown. However, with the ELPI+, the operation principle does not fundamentally enable utilisation of size-dependent effective density (see 2.3), which should be acknowledged. Still, this uncertainty of ELPI+ can be estimated by comparing the density-corrected results to the DMPS/SMPS results which are not as vulnerable to errors in terms of varying effective density.*"
Also, in Methods -section (2.3):
"*Also, with the ELPI+, data analysis with size-dependent effective density is not straightforward due to the cascade impactor measurement.*"

A sensitivity analysis showing how variations in effective density across different size ranges impact LDSA calculations would strengthen the argument.

- Thank you for this suggestion. Please, see the response to the previous comment.

The significant variation between ELPI+ and SMPS size distributions in Prague, especially for accumulation mode particles, warrants a more detailed explanation, considering factors beyond effective density.

- In Figure 2, it can be seen that the differences between ELPI+ and SMPS are much less significant after the density correction for the data, so it is reasonable to consider that it is the main reason for the difference in the results without any corrections.

- One additional reason is the varying operation principles: SMPS measures particles based on their number concentration (CPC), and then LDSA[al] is calculated by assuming spherical particle shape (which of course is an approximation as explained in the methods section). ELPI+ (also Partector), however, measures concentrations based on the particle charge, and, therefore, particle shape also affects the measured concentrations. Especially with larger particles, agglomerated structures can cause considerable differences between the detection methods of a CPC and electric current, which is also discussed in the method sections. Even though some device-related uncertainties can slightly affect the obtained results, these two points are most likely the main reason behind the difference.

- These explanations are discussed thoroughly in the manuscript, but we added reference to section 2.1.1. in the text, to help the reader to find further explanations for the mentioned behaviour of the size distributions.

Line 359: Quantitative analysis should be provided to support the conclusion about decreased differences after peff-correction.

- By comparing size distributions in Figure 1 and 2, it can be seen that the difference between ELPI+ and DMPS/SMPS is clearly less significant after the density correction. Also, Figure 4 and (original Figure S13, new Fig S24) support this idea in terms of total concentrations. The data behind the figures is provided in original manuscript Table S1 (new S2), and also main numbers of the results have been provided in Section 3.2.2. Thus, we believe that it is rather evident, that the differences between the methods decreased after the effective density correction. It is not clear what kind of additional quantitative analysis is requested in this comment. It is also worth considering, that LDSA[al] has not strictly defined reference measurement method, and all the common LDSA[al] methods have both strength and weaknesses (as shown in the manuscript). Therefore, it is not straightforward to quantitatively state which methods are the most correct, nor the exact effect of different corrections, as there is no reference method available. One main point of this study is to show how the results with different methods vary in different conditions and what factors influence the obtained results, which is crucial when comparing LDSA[al] results with different instrumentation, and not yet well understood. However, additional discussion related to the effective density correction is added in the revised text (see previous comments) to help readers to evaluate the effect of corrections.

- Also, we added hourly averaged scatter plots of the Helsinki and Prague data, which help to analyse the differences after the corrections (Fig S16-18): **"Also, the differences in the scatter plot analysis with the hourly averaged data mainly decreased after the correction (Fig. S16-18)."**

Figure 3: The opposite trends of overestimation and underestimation for ELPI+ and SMPS require more in-depth discussion.

- Thank you for this suggestion. In the revised text, discussion about these has been added: ***The overestimation of total LDSA$^{al}$ concentration of the ELPI+ with the standard $\rho_{eff}$ assumption (Fig. 3) can be explained with the conversion from electric current into to LDSA$^{al}$, as the calculation considers the particles to have larger mobility equivalent size than they have in reality, causing the conversion factors into to LDSA$^{al}$ to be too high (see also Lepistö et al. 2020)***. *As seen, the majority of LDSA$^{al}$ concentration in the studied sites was attributable to particles smaller than 500 nm (mobility equivalent diameter). Thus, the DMPS, SMPS and Partector are less vulnerable to errors related to the effective density if the concentration of particles larger than 500 nm is not high* ***as both the measurement (charging efficiency and size classification) and lung deposition efficiency are dependent on the same quantity (mobility equivalent size). The slight underestimation in Fig. 3, is related to the concentrations of particles larger than about 500 nm, where the dominant deposition method changes from diffusion to impaction, causing the DMPS and SMPS to underestimate the deposition efficiency.***"

Lines 408-412: The statement about hygroscopicity-corrected size distributions for SMPS and DMPS should be reconsidered:

The changes before and after corrections may be more related to particle size distribution than hygroscopicity or chemical composition.

- The hygroscopicity correction does not change the measured particle size distribution. The correction affects only the estimated particle lung deposition efficiency function (see Figure S7). Thus, the changes after the hygroscopicity correction are related to different particle lung deposition efficiency caused by the hygroscopic growth of particles in the respiratory system. Therefore, in all three cases (1. general assumptions, 2. effective density corrected, and 3. effective density & hygroscopicity corrected), the input size distribution is always the same. This point is clarified in the revised Methods - section: ***"It should be noted, that the hygroscopicity correction only changes the estimated lung deposition efficiency of particles, not the initial size distribution or the surface area of the inhaled particles."***

A more detailed analysis of how the PSD itself influences the observed changes after hygroscopicity correction is needed.

- Please, see our response to the previous comment. The PSD itself does not change due to the correction. But, of course, the initial particle size distribution affects how much the hygroscopicity-correction changes the result. This is explained in the revised manuscript: "***In general, hygroscopicity correction decreased the lung deposition***

*efficiency of particles smaller than 200 nm whereas it increased the deposition efficiency of particles larger than 200–400 nm (see also Fig. S7)."*

- Also, the discussion in Section 3.2.3 was slightly modified:
*"...With the ELPI+, LDSA$^{al}$ concentration with the general assumptions was 107–114 % of the hygroscopicity-corrected result in all the cases. With the DMPS and SMPS, LDSA$^{al}$ with general assumptions was 95–104 % of the ones with the hygroscopicity-correction.* **This result can be explained due to the balancing effects of particles smaller than 200 nm and larger than 200–400 nm in terms of the hygroscopicity correction.** *Thus, by a coincidence, accuracy of the absolute LDSA$^{al}$ concentration measurement was not significantly affected due to the particle hygroscopicity.* **However, it's worth noting that this result may depend on the location and urban environment. For example, high concentration of accumulation mode particles can potentially cause underestimation of LDSA$^{al}$ without hygroscopicity correction.** *Also, it's important to note that the hygroscopicity correction still affected the relationship between the studied instruments (Fig. 6)."*

The relative importance of hygroscopicity versus PSD in determining the final LDSA values should be discussed.

- Please, see the responses to previous comments.

Stronger quantitative support is needed for the conclusion about the agreement between methods after hygroscopicity correction.

- Similarly, as with the response regarding the effective density, the lack of reference instrument for LDSA$^{al}$ challenges quantitative analysis of the strengths and weaknesses between the different methods. One of the main points of this study is to show how different factors like the effective density or hygroscopicity can affect the accuracy of different LDSA$^{al}$ instruments in ambient conditions, which is not well understood currently. In Fig. 5 and 6, the changes after hygroscopicity correction can be observed (see also Table S2 and Fig. S21). The hygroscopicity correction decreases the lung deposition efficiency of particles smaller than 200 nm, whereas it increases the deposition efficiency of larger accumulation mode particles (see also Fig. 5). Then, the differences between instruments are discussed in terms of total concentration. It is not clear what kind of additional quantitative analysis is requested in this comment.
- However, we added new scatter plots of hourly averaged data (as mentioned, Fig. S16-18): *"Thus, by a coincidence, accuracy of the absolute LDSA$^{al}$ concentration measurement was not significantly affected due to the particle hygroscopicity* **which can also be seen in the hourly averaged scatter plots (Fig S16-18)."**

Line 497: The conclusion about neglected hygroscopicity not considerably changing the results due to balancing effects should be presented more cautiously, acknowledging that it may only be valid under certain conditions.

- We agree, this point is now acknowledged (see responses to previous comments).

**Technical Corrections:**

Line 312: Add the Wu et al. (2023) reference to the reference list.

- Thank you for pointing this out. We have added the reference to the list.

Check for consistency in terminology throughout the manuscript, particularly in the use of LDSA and LDSAal.

- In the text, LDSA$^{al}$ notation is used thoroughly. As mentioned in an earlier response, this notation is used to clarify that the results consider alveolar deposition, and this clarification is also mentioned in the revised text. LDSA without "al" is used only once: "*It's worth noting that, in different studies, LDSA can be referred also when considering other respiratory regions than only alveoli (e.g. Liu et al. 2023). Here, the notation LDSA$^{al}$ is used to clarify that only alveolar deposition is considered in this study.*"

**Author response to Reviewer #2**

The manuscript describes a comprehensive study on the measurement of the alveolar lung deposited surface area (LDSA) concentration at different urban locations. LDSA is generally considered to be a more health relevant concentration metric than the usually measured mass or number concentrations. Since it can readily and relatively cheaply be measured by unipolar diffusion charging of particles, followed by a measurement of the current caused by the particle-borne charges, it has raised increased interest over the last years, among others in the long-term monitoring of atmospheric particles.

In the present study, LDSA concentrations were measured by using both an electrical diffusion charger device (Partector), which delivers LDSA concentrations directly as well as size distribution measurement devices (ELPI, SMPS, DMPS), which were used to calculate the LDSA concentrations from the measured size distributions. Due to concurrent measurements of the number size distributions based on the aerodynamic and mobility diameter, the authors obtained information on the effective density of the particles and used this to correct the measured data. Further, the authors investigated to what extent particle hygroscopicity may affect LDSA measurements. Measurement uncertainties caused by these two properties have thus far not been considered and provide useful added value to the scientific literature.

While the manuscript is well-written and timely, it unfortunately suffers from several shortcomings. I suggest that the manuscript should undergo a major revision before it can be accepted for publication.

- We thank you for giving these valuable comments on the manuscript. We certainly agree that, in the scientific literature, measurement uncertainties related to the effective density and hygroscopic growth (and to other factors as well) are not well understood and they are usually missing in studies focusing on the LDSA. That's why we believe that this manuscript would provide useful and novel information for the scientific community, which again could further stimulate discussions on the ways in which LDSA is utilised as a monitored metric. Also, we hope that our study helps the interpretation of measurement uncertainties related to different LDSA measurement methods, which are not commonly disclosed in LDSA related studies. We have carefully considered your comments on the manuscript, and our responses (and changes) are provided below. We hope that our responses and changes are satisfactory.

**Main criticism:**

The authors correctly point out that the LDSA-metric suffers from the lack of a proper definition. The lung deposition efficiency depends on various breathing parameters, as well as age and sex of an individual. Furthermore, different models as well as approximations to the model results exist. To the best of my knowledge, the only device, for which all parameters have been fully disclosed, is the TSI NSAM, which has been calibrated to mimic the breathing parameters of a "reference worker", applied in the ICRP model (Fissan et al., 2007). Here, the authors applied different breathing parameters to an approximation of the same model, published in the Hinds textbook. However, the textbook claims that this approximation is only accurate to within +/- 0.03, which means that the approximation is particularly inaccurate in the important size range of the accumulation mode, where the deposition efficiencies are rather low. Why has this approximation still been used? Why not the KDEP computer model, which is a free software, that calculates the values of the ICRP model much more accurately?

- Thank you for this important comment. As mentioned, LDSA lacks proper definitions which challenges the comparison of results from different studies and instruments. This challenge affects also this study because the exact lung deposition parameters used in the calibration of Partector are not disclosed, at least in the paper by Fierz er al. 2014 or in the device manual. The reason to use the Hinds approach comes from the ELPI+ LDSA calibration, which has been done based on those equations (Lepistö et al. 2020). Therefore, to reduce the uncertainty between the size distribution methods, the DMPS/SMPS results have been calculated by utilising the same approach. Therefore, the only uncertainty related to lung deposition efficiencies in the comparisons is the Partector calibration, which likely varies from the size distribution devices. Thus, we think that this approach is reasonable. We acknowledge the challenges related to the approximations, but on the other hand, as LDSA lacks proper definition, "the most proper" lung deposition function cannot be chosen because all approaches have their own problems. The advantage of the Hinds' approach is that the functions are well defined, so they are easy to use for all, and also, they are based on average lung deposition considering both male and female at three different exercise levels. Even though more sophisticated computer models could be used, it should be noted that there are many different models available (e.g., ICRP, MPPD, IDEAL ) which have differences especially with certain input parameters. So, overall, we believe that the regardless of the utilised lung deposition function, there are always uncertainties related to the chosen method in LDSA studies. That is also why we think that it is necessary to create a proper definition for LDSA, so that all device manufacturers/scientist could use the same deposition model (in monitoring measurements).

- In general, we think that this comment highlights the main problem related to LDSA as a monitored metric. Even though, this problem is somewhat understood in the scientific community, we feel that this discussion is however missing in the current LDSA related literature. One main point of this article is to disclose the challenges related to the current LDSA measurement methods, and how complicated the measurement actually is, especially, when considering the effective density and hygroscopicity. Therefore, we believe that this topic actually is one of the main strengths of this manuscript, even though, it causes also unavoidable challenges in the data analysis.

- Already in the original manuscript, we have discussed the challenges of LDSA measurement, especially in terms of effective density and hygroscopicity. Thanks to this

comment, we have added more comprehensive discussion related to the challenges of the measurement, covering both the effect of the chosen lung deposition model, and its parameters (e.g., the breathing pattern) as well as the relevance of surface area (e.g., compared to mass) with the larger accumulation mode particles, or soluble particles, for which also the challenges in LDSA measurement start to increase. These topics were shortly mentioned in the original text, but in the revised version, these topics and their role are better disclosed. See our responses to later comments to see these changes.

How were the breathing parameters for the approximation chosen?

- The breathing parameters have been chosen based on the ICRP-model parameters for adult male and female with three different exercise levels (sitting, light exercise, heavy exercise). In the revised manuscripts, the parameters are shown in the new Table S1.

On another note: the relatively new draft CEN/TS 18073 finally defines a convention for the breathing parameters and lung deposition values to be applied in LDSA devices of the future.

- This is a great step forward. In terms of this manuscript, we think that this new draft cannot however to be taken into account yet, as both Partector and ELPI+ have already their own calibration principles, so at this stage, the new definition (which is also only a draft yet), cannot be applied in the results. However, the need to define common practices for LDSA measurement has been mentioned already in the original manuscript (Strengths and limitations)*: "Still, it would be beneficial to determine common practices for LDSA[al] measurement in general regarding the utilised lung deposition function".*

- To emphasise this topic, we mention this now also in the conclusions: ***"Still, further research on the relevance of surface area in terms of larger and soluble particles as well as determination of common practises for LDSA[al] measurement related to the utilised deposition model and its parameters are needed to better understand the relevance and improve the suitability of LDSA[al] in terms of air quality monitoring."***

It is not clear to me, how the correction for the effective density was applied. Only to transfer from aerodynamic to mobility diameter and then calculate the surface area (and lung deposition efficiency?) based on this value? The original ICRP model allows for differentiating between the diffusional deposition, for which the mobility diameter would be of relevance, and the mass-driven deposition, for which the aerodynamic diameter is of relevance. Has this difference been considered?

- The effective density correction considers both aerodynamic and diffusional deposition separately. The diffusional deposition is calculated based on the mobility diameter and the inertial deposition is based on the aerodynamic diameter, similarly as in the original ICRP-model. So yes, the density correction affects both the surface area and the lung deposition efficiency.
- We have clarified the effective density correction in the revised text: ***"The correction calculates the density corrected deposition function by considering inertial deposition (aerodynamic diameter) and diffusional deposition (mobility equivalent diameter) separately (see ICRP 1994, Lepistö et al. 2020)."***

Similarly, I wonder how the hygroscopicity correction has been applied. Which rh has been assumed? 100% as in the respiratory tract? Was the grown diameter only been used to determine the lung deposition or also the surface area?

- The hygroscopicity correction is based on the method by Vu et al. (2015), which is also cited in the manuscript. Vu et al. collects data of particle hygroscopic growth measured in different urban areas and countries. The original data has typically been measured with RH around 90 %, but Vu et al represent a method how to calculate the hygroscopic growth in the lung (RH = 99.5%) based on this data, and how this changes the lung deposition efficiency. The hygroscopicity correction only changes the lung deposition efficiency, not the surface area.

- This is clarified in the revised manuscript: "***The growth rate is calculated by assuming relative humidity of 99.5 % in the human lung. It should be noted, that the hygroscopicity correction only changes the estimated lung deposition efficiency of particles, not the initial size distribution or the surface area of the inhaled particles***."

In fact, I have been wondering for a relatively long time, how relevant the effect of hygroscopicity is in general for the relevant surface area. All the relevant toxicological studies that I know have shown that surface area is a good predictor for health outcomes of non-soluble particles only, whereas soluble particles would dissolve in the lung and consequently their mass would be the relevant metric. Would thus the surface area of a hygroscopically grown particle be of relevance?

- This topic is very interesting and important when considering LDSA as a metric for adverse health effects. The hygroscopic growth clearly changes the lung deposition efficiency especially with the larger accumulation mode particles. But, as mentioned, it is not evident whether surface area is the most relevant metric with these particles anymore (compared e.g. to mass). With ultrafine particles, the importance of surface area has been observed in many studies. This topic was briefly discussed already in the original manuscript (section 3.2.5), where the relevance of surface area compared to mass was questioned in terms of larger accumulation mode particles and secondary aerosol. However, soluble particles were not mentioned, which of course is very important as well.

- Changes in the revised text (section 3.2.5): *"In terms of particle health effects, the relevance of surface area is likely the highest with the smaller ultrafine and soot particles (Oberdorster 2005, Schmid and Stoger 2016, Hakkarainen et al. 2022) whereas with, larger particles, secondary aerosol **and soluble particles,** the health effects have been strongly associated with the mass concentration (e.g., Lakey et al. 2016, Lin et al. 2016, **Yu et al. 2022**, Yang et al. 2023). **Therefore, it is uncertain whether surface area deposition is relevant in terms of larger and soluble particles**. Thus, in terms of monitoring the effects of nearby local pollution sources in a dense air quality monitoring network, LDSA[al] should be well a suitable and potential metric in terms of the particle health effects…"*

    Yu, W., J. Chen, W. Qin, M. Ahmad, Y. Zhang, Y. Sun, K. Xin, J. Ai. Oxidative potential associated with water-soluble components of PM2.5 in Beijing: The important role of anthropogenic organic aerosols, Journal of Hazardous Materials, 433, 2022. https://doi.org/10.1016/j.jhazmat.2022.128839.

- Also, in Strengths and limitations: *"The results suggests that the main challenges of the measurement start to have a considerable effect on the results only with high concentrations of accumulation mode particles larger than 200–400 nm. **Also, the relevance of surface area in terms of the adverse health effects is not as evident anymore with larger and soluble particles compared to solid ultrafine particles or soot.** Therefore, LDSA[al] could be a suitable parameter for detecting the spatial differences in the particulate pollution within cities as the effects of nearby pollution sources, like traffic, are commonly observed with ultrafine and soot particles that are smaller than 200 nm."*

- Also, in the Conclusions: *"However, the challenges of the measurement were mainly related to the accumulation mode particles larger than 200–400 nm, **for which surface area may not be as relevant in terms of the adverse health effects as with smaller ultrafine particles or soot**. Therefore, regardless of the method, LDSA[al] should be well suitable when considering its utilisation **to detect the effects of nearby local pollution sources** in dense air quality monitoring networks as long as the effects of larger particles are addressed either by removing them from the sample or also measuring the regional background concentration. **Still, further research on the relevance of surface area in terms of larger and soluble particles as well determination of common practises for LDSA[al] measurement related to the utilised deposition model and its parameters are needed to better understand the relevance and improve the suitability of LDSA[al] in terms of air quality monitoring**."*

The comparisons of the LDSA values are only based on averages and 25th/75th percentiles, which hides a lot of information. I think it would be more informative, if the data (e.g. 1 h averages) are presented as scatter plot diagrams.

- Thank you for the suggestion. We agree that scatter plot comparisons would be beneficial for the article. We added hourly averaged scatter plots for the Helsinki and Prague data (including Partector and DMPS/SMPS) (Supplementary Fig. S16-18). Also, we added scatter plots of minute averaged results from all the sites included in the ELPI+ and Partector comparison  (Supplementary figures S20-22). These plots are also shortly mentioned in the analysis:
  3.2.2**: "Also, the differences in the scatter plot analysis with the hourly averaged data mainly decreased after the correction (Fig. S16-18)."**
  3.2.3**: *"Thus, by a coincidence, accuracy of the absolute LDSA[al] concentration measurement was not significantly affected due to the particle hygroscopicity **which can also be seen in the hourly averaged scatter plots (Fig S16-18)."***
  3.2.4: **"Despite the underestimation of the Partector in sites with high $PM_{2.5}$ and low PN, the correlation of the measured concentrations with the ELPI+ and Partector was still strong in all the studied sites ($R^2$: 0.85-0.98, Fig. S20-22), showing that the challenges are mainly related to the utilised calibration factor. "**

- In addition, to be more transparent with the utilised data, similar histogram figures as with PM, BC, NO, PN are given with the LDSA (all instruments) in the supplementary of the revised manuscript (Fig S12-15).

Why are the BC and NO measurements mentioned in the text and data shown on Table 1, but not further dicussed? Either add a discussion (if they provide a benefit to this study) or remove any text on these two metrics.

- BC and NO data are shown because they give information about the measurement sites and the dominating emission sources. Both are good tracers for the nearby traffic. In addition, BC is a tracer for residential wood combustion, which is especially relevant in Finland during winter. Therefore, we think that the discussion in the beginning of section 3.1 can be better justified if BC and NO data are given in addition to only PM and PN.

- In the revised manuscript, the discussion in the beginning of section 3.1. was slightly modified to highlight the differences between the studied sites and episodes better: *"In general, the contribution of the nearby road traffic was clearer in Helsinki than in Prague* ***due to the shorter distance from the passing vehicles to the measurement site, partly explaining the relatively higher average PN, NO and BC concentrations compared to PM$_{2.5}$.****"* Also:
  *"In Helsinki, PM$_{2.5}$ concentration increased during the inversion- and LRT-episodes. During the inversion episode also PN, NO and BC concentrations increased considerably which indicates local contribution*. ***In addition to road traffic, higher BC during the inversion episode indicates effects of residential wood combustion, which is typical emission source in Finland during winter (e.g. Teinilä et al. 2022)."***

Each of the measurement campaigns has been relatively short (few weeks at most) and all were during winter/early spring. I therefore wonder, how representative the data can be. A disclaimer should be added, mentioning this shortcoming and that more research over longer periods and covering different seasons is needed.

- We agree that more research over longer periods, including different geographic regions, urban areas, and seasons are needed in the future. However, it should be noted that the measurements in Finland were conducted during winter whereas the measurements in Central Europe were done during spring. The weather conditions between the Finnish winter and Central European spring are very different, especially in terms of sun light and temperature, which affect also rain/snow fall and boundary level height. Therefore, we think that the study covers clearly two different seasons.

- In the revised text, the ending of "Strengths and limitations" section was modified: *"In this study, the detailed comparison was, however, only done for road traffic environments in lowly or moderately polluted regions.* ***Also, the measurements of this study were rather short, and conducted only during certain seasons.*** *Thus, there is still a need for studies of particle effective density, hygroscopic growth, and particle size distributions along with LDSA[al] measurement in different urban environments and in highly polluted regions,* ***including long-term data,*** *to better understand the universal suitability and behaviour of the metric."*

**Specific points:**

Line 93/94: The electrical particle sensors do not need to assume a size distribution due to the similarity of the charging efficiency and the lung deposited surface area per particle (see e.g. Todea et al., 2015), which only holds in a size range from approximately 20 to 400 nm. Note that the analysis in this paper is based on the ICRP model with the breathing parameters used by TSI for NSAM.

- In principle, the size distribution needs to be assumed, as the sensors are only accurate in the mentioned particle size range. As the sensors do not remove the larger particles from the sample, the concentrations of larger particles needs to be assumed low. But technically, it is true that the sensors do not utilise size distribution assumptions in their calculation, and this has been clarified in the revised manuscript:

  In the methods: *"The Partector first charges the sampled particles in a diffusion charger and then converts the detected electric current caused by the sampled particles into LDSA$^{al}$ concentration with a single calibration factor. **The chosen calibration factor is the response coefficient between the electric current and LDSA$^{al}$ at 100 nm, which typically is close to the peak size of LDSA$^{al}$ size distributions in urban environments (Fierz et al. 2014).**"*

- Regarding the referred lines, the word "distribution" was removed: "assuming certain particle size in the calibration".

Line 100 ff.: Whereas the effect of hygroscopicity and effective density on the lung deposition are extensively discussed here, a discussion on the effect of (individual) breathing parameters is missing.

- As the main point of this study is not to compare the effects of the exposed population (anatomy, activity, breathing pattern), this topic is not deeply discussed in the manuscript, as the main focus is on the particle-related effects (effective density, hygroscopicity). However, it is important to acknowledge that particle lung deposition is always dependent on the individual, and this topic is further elaborated in the revised text:

- In the Introduction: *"**In addition, particle lung deposition efficiencies are individual and dependent on the human anatomy and the breathing pattern, and, thus, approximations of the exposed population are always needed.** "*

- In 2.1.1.: *"...On the other hand, the neglected hygroscopic growth of particles, together with the standard density assumption, are often the only reasonable options for monitoring measurements as the consideration of these parameters require additional sophisticated instrumentation. **Also, in principle, particle lung deposition efficiencies are individual and dependent on the human anatomy and the breathing pattern. Thus, the utilised lung deposition efficiency functions in device calibrations are always approximations, and the chosen approaches may vary with different instruments**. Thus, in addition to the uncertainties between the different operation principles of the methods, LDSA$^{al}$ measurements also have uncertainty in the estimation of the actual particle lung deposition."*

- Also, in Strengths and limitations: *"Also, further studies on the effects of the chosen lung deposition model, and its parameters, in terms of LDSA[al] measurement with different devices would be beneficial to better understand how well the current measurement methods represent actual lung deposition, in addition to the effects of effective density and hygroscopic growth."*

- Conclusions: *"Still, further research on the relevance of surface area in terms of larger and soluble particles as well determination of common practises for LDSA[al] measurement related to the utilised deposition model and its parameters are needed to better understand the relevance and improve the suitability of LDSA[al] in terms of air quality monitoring."*

Line 140 ff.: Did you try to use the total current (e.g. in size range 20-400 nm), measured by ELPI directly to determine the LDSA concentration instead of calculating it from the size distribution? Since ELPI uses a unipolar diffusion charger with in principle similar charging characteristic as the Partector (I assume), this may directly yield the LDSA concentration.

- In this manuscript, the suggested "single-factor" LDSA calibration for ELPI+ is not considered, but an detailed comparison of that method against the used "better" stage-specific calibration is presented in Lepistö et al. (2020), which is also cited in the manuscript. Also, Lepistö et al. (2023) (also cited) compared LDSA size distributions measured with the ELPI+ in different urban environments and geographic regions. In that study, also the "single-factor" LDSA calculation with ELPI+ is compared to the stage-specific method in terms of total concentration. The single-factor calibration has also been utilised e.g., by Kuuluvainen et al. (2016) and Pirjola et al (2017), before the stage-specific calibration was introduced (Lepistö et al 2020). In general, the single-factor LDSA calibration with the ELPI+ includes the same fundamental limitation with the particles larger than 400 nm, but it is also slightly less accurate with particles in the "suitable" size range of 20 – 400 nm (see Lepistö et al. 2020 and 2023). Note that the utilised stage-specific calibration also converts the measured electric current directly to LDSA, it just has 14 calibration factors (for each stage), in comparison to only one like with the sensors.

  Kuuluvainen, H., Rönkkö, T., Järvinen, A., Saari, S., Karjalainen, P., Lähde, T., Pirjola, L., Niemi, J.V., Hillamo, R. & Keskinen, J. (2016). Lung deposited surface area size distributions of particulate matter in different urban areas, Atmospheric Environment, Vol. 136 pp. 105-113. https://www.sciencedirect.com/science/article/pii/S1352231016303016.

  Pirjola, L., J. V. Niemi, S. Saarikoski, M. Aurela, J. Enroth, S. Carbone, K. Saarnio, et al. 2017. Physical and chemical characterization of urban winter-time aerosols by mobile measurements in Helsinki, Finland. Atmos. Environ. 158:60–75.

Line 171: "...converted from the number concentration" should read "...converted from the number size distribution".

- Thank you for pointing this out. We have corrected the sentence in the revised manuscript.

Line 173: The Partector does not assume a size distribution (see above).

- As mentioned in the response to the earlier comment, the calibration method of Partector has been clarified in the revised manuscript.

Line 302/303/Figures S8-S11: Why are these graphs termed "deviation" plots? Aren't these simply histograms, showing how often (relatively) certain concentration ranges occurred?

- In the revised manuscript, the term "deviation plot" have been changed to "histograms".

Line 465/Figure 7: It is striking that the Partector/ELPI ratio is always below 1, showing that the difference must be systematic

- We agree. In the revised text, this observation is brought up in the revised text:

- In 3.2.4.: *"On the other hand, in all the studied sites, the Partector measured lower concentrations than the ELPI+ (also after density corrections in Helsinki and Prague), suggesting systematic difference between the methods, which may e.g., be related to different lung deposition models in the device calibrations."*

- Also, same section: *"Despite the underestimation of the Partector in sites with high PM$_{2.5}$ and low PN, the correlation of the measured concentrations with the ELPI+ and Partector was still strong in all the studied sites (R$^2$: 0.85-0.98, Fig. S20-22), showing that the challenges are mainly related to the utilised calibration factor. "*

- In Strengths and limitations: *"However, for example, in Fig. 7, Partector systematically measured lower concentrations than ELPI+ in all the studied sites, which suggest that varying deposition models in the devices' calibration could also have had an effect on the results."*